# Ultrastrong nanocrystalline steel with exceptional thermal stability and radiation tolerance

Congcong Du[1], Shenbao Jin[2], Yuan Fang[3], Jin Li[4], Shenyang Hu[5], Tingting Yang[1], Ying Zhang[1], Jianyu Huang[1], Gang Sha[2], Yugang Wang[3], Zhongxia Shang[4], Xinghang Zhang [4], Baoru Sun[1], Shengwei Xin[1] & Tongde Shen [1]

Nanocrystalline (NC) metals are stronger and more radiation-tolerant than their coarse-grained (CG) counterparts, but they often suffer from poor thermal stability as nanograins coarsen significantly when heated to 0.3 to 0.5 of their melting temperature ($T_m$). Here, we report an NC austenitic stainless steel (NC-SS) containing 1 at% lanthanum with an average grain size of 45 nm and an ultrahigh yield strength of ~2.5 GPa that exhibits exceptional thermal stability up to 1000 °C (0.75 $T_m$). In-situ irradiation to 40 dpa at 450 °C and ex-situ irradiation to 108 dpa at 600 °C produce neither significant grain growth nor void swelling, in contrast to significant void swelling of CG-SS at similar doses. This thermal stability is due to segregation of elemental lanthanum and (La, O, Si)-rich nanoprecipitates at grain boundaries. Microstructure dependent cluster dynamics show grain boundary sinks effectively reduce steady-state vacancy concentrations to suppress void swelling upon irradiation.

[1] Clean Nano Energy Center, State Key Laboratory of Metastable Materials Science and Technology, Yanshan University, Qinhuangdao 066004, China. [2] Department of Materials Science and Engineering, Nanjing University of Science and Technology, Nanjing 210094, China. [3] State Key Laboratory of Nuclear Physics and Technology, Center for Applied Physics and Technology, Peking University, Beijing 100871, China. [4] School of Materials Engineering, Purdue University, West Lafayette, IN 47907, USA. [5] Pacific Northwest National Laboratory, P.O. Box 999, Richland, WA 99352, USA. These authors contributed equally: Congcong Du, Shenbao Jin, Yuan Fang, Jin Li. Correspondence and requests for materials should be addressed to G.S. (email: gang.sha@njust.edu.cn) or to Y.W. (email: ygwang@pku.edu.cn) or to T.S. (email: tdshen@ysu.edu.cn)

Stainless steels (SSs) have numerous applications in the automotive[1], construction[2], and nuclear power industries, with the global production of stainless and heat resisting steel in 2017 reaching 48,081,000 metric tons[3]. High-strength SSs not only have significant economic and environmental impact, but they also see a variety of applications in extreme environments. However, conventional austenitic SSs have relatively low strengths. For example, 304-type SSs have a yield strength of ~230 MPa[2]. Their mechanical strength can be further enhanced by grain refinement based on the Hall–Petch relationship[4,5], i.e., the yield strength of a metal is inversely proportional to the square root of its grain size. NC metals composed of nanograins (with grain sizes below 100 nm) are predicted to be much stronger than their CG counterparts. However, NC metals often suffer from prominent grain growth between 0.3 and 0.5 $T_m$[6]. This tendency of grain coarsening becomes a major challenge for producing bulk NC metals by consolidating NC powders. Kinetic and thermodynamic strategies are often used to increase the thermal stability of NC materials. In the kinetic approach, grain boundaries (GBs) of the NC materials are pinned in various ways —such as solute drag[7], second-phase particle pinning (Zener pinning)[7,8], chemical ordering[9], or porosity[10]—to decrease GB mobility. In the thermodynamic approach, the driving force for grain growth is curtailed by reducing the specific GB energy. The thermodynamic approach, often achieved by the GB segregation of solutes and the resultant decrease in GB energy, has been theoretically analyzed[11–13] and experimentally studied in such NC materials as Al[14], Cu[15], Fe[16], and Fe–Cr alloys[17]. Recent studies show that nanograins can be stabilized by the nanoscale chemical distribution in NC W–Ti alloys[18] or by an autonomous grain boundary evolution to low-energy states due to the activation of partial dislocations in the plastic deformation of pure NC Cu and Ni[19].

Structural materials for next-generation fission and fusion reactors must be stable at very high temperatures (up to 1000 °C) and should be able to withstand intense irradiation (up to 200 dpa)[20]. Ferritic steels are widely used in nuclear reactors due to their excellent void swelling resistance, but their creep resistance is poor due to their body-centered cubic structure. Face-centered cubic austenitic SSs have high creep resistance. However, conventional austenitic CG-SSs exhibit poor void swelling resistance in comparison with ferritic steels[21,22]. Engineering design of reactor structures requires a material with a swelling level of less than 5%[23]. Unfortunately, austenitic SSs often experience void swelling levels as large as several tens of percent[21,22,24–29]. Designing austenitic SSs with excellent swelling resistance against intense irradiation is thus a challenge to the nuclear materials community.

Introducing a high density of sinks, such as GBs[30–33], interfaces[34–38], and nanoprecipitates (NPs)[39,40], is an effective way to enhance the radiation tolerance of materials. In particular, nanostructured materials, such as nanotwinned metals[33,41], nanolaminates[33,42], nanocrystalline metals[33,43], etc., with unique predesigned defect sinks have the potential to provide both high strength and radiation resistance. Among them, oxide dispersion strengthened (ODS) nanostructured ferritic alloys (NFAs)[44] have been extensively studied for potential nuclear applications. The NFAs are composed of oxide NPs, several nanometers in diameters, dispersed in ultrafine-grained (UFG, typically 200 nm to 1 μm[45,46]) matrix. The NFAs are often produced by mechanically alloying pre-alloyed or elemental powder and $Y_2O_3$ nanoparticles (~20 nm), and subsequently consolidating the mechanically alloyed powers at high temperatures. During the mechanical alloying (MA), $Y_2O_3$ is dissolved into the steel matrix. The subsequent consolidation leads to the formation of (Y–Ti–O) NPs, which have a high number density of ~$10^{23}$ to $10^{24}$ $m^{-3}$, located

in grain interior and on GBs[47]. The (Y–Ti–O) NPs/matrix interfaces and GBs act as effective sinks for irradiation-induced defects. Such GB sinks also operate in austenitic SSs, because a decrease in grain size from 50 μm to 450 nm has been reported to be effective in reducing the void swelling[48]. Recent results indicate that the void swelling of austenitic UFG-SS with a grain size of 100 nm is nearly an order of magnitude smaller than that of CG-SS[49]. However, significant scientific questions remain unanswered, such as whether we design thermally stable bulk NC-SS, and whether the radiation stability and mechanical behavior of NC-SS be superior to those of CG- and UFG-SS.

In this research, we develop a powder metallurgy approach to prepare NC supersaturated Fe(Cr,Ni,La) solid solution alloy powders by MA. We then consolidate the NC powders to form a bulk NC-SS via consolidation at a high temperature of 1000 °C under a high pressure of 4 GPa. The dissolved La solutes segregate to GBs and result in outstanding thermal stability of nanograins up to 1000 °C (0.75 $T_m$). The NC-SS has an ultrahigh yield strength of ~2.5 GPa and a total strain of ~0.4 under compression. Moreover, the NC-SS exhibits no void swelling upon high temperature and high dose irradiations. Simulations by using a microstructure dependent cluster dynamics model suggest that the ample GB sinks in the NC-SS effectively lower the steady-state vacancy concentrations upon irradiating, and thus eliminate void swelling.

## Results

**Phase constituents**. As-received CG-SS powder is composed of an austenitic phase (Supplementary Figure 1). After mechanically alloying the CG-SS and 1 at% elemental La powders, the resulted NC-SS powder contains both austenitic and martensitic phases. This is understandable since deformation can trigger an austenite-to-martensite transformation. No diffraction peaks of elemental La, within the detection limit of the X-ray diffraction, are observed in the NC-SS after MA, suggesting that elemental La is incorporated into the lattice of SS matrix by MA. Since the room-temperature solubility of La in Fe is negligible, the NC-SS after MA should be a supersaturated solid solution. The formation of a supersaturated solution can be further confirmed by comparing the lattice parameter of NC-SS containing 1 at% La, 0.35933 ± 0.00012 nm, with that of NC-SS without La, 0.35827 ± 0.00022 nm. In addition, note that MA technique often introduces numerous lattice defects such as excess vacancies, dislocations, and GBs, etc. Thus, La may not be solely dissolved in the bulk lattice but could also be segregated at these lattice defects to a large extent. After consolidating the mechanically alloyed NC-SS powder containing 1 at% La, a bulk NC-SS is formed and composed of only austenitic phase. For clarity, we use NC-SS to represent the consolidated bulk NC-SS containing 1 at% La in the following descriptions.

**Mechanical properties**. Figure 1a shows the compressive and tensile stress–strain curves of consolidated NC-SS bulk. The yield strength in both compression and tension is 2.5 ± 0.4 GPa. This yield strength is ten times that (230 MPa) of CG-SS[2], much stronger than that (0.85–1.35 GPa) of NFAs[44], and superior to those (1.95–2.21 GPa) of a recently developed maraging steel with NPs (MS-NPs)[50] and Mn steel with high-density dislocations (MS-HDDs)[51], as shown in Fig. 1b. These results suggest that the NC-SS should be one of the strongest crystalline steels. Additionally, the NC-SS under compression exhibits a large fracture strain of ~0.4, indicative of the capability of plastic deformation under compression. In contrast, the NC-SS under tension exhibits rapid necking after its tensile strength reaches 2.9 GPa, leading to a fracture strain of ~0.04. Note that the compressive curve of the NC-SS displays a softening after yielding, suggesting that there are damages and/or cracks developed. The morphology of the top

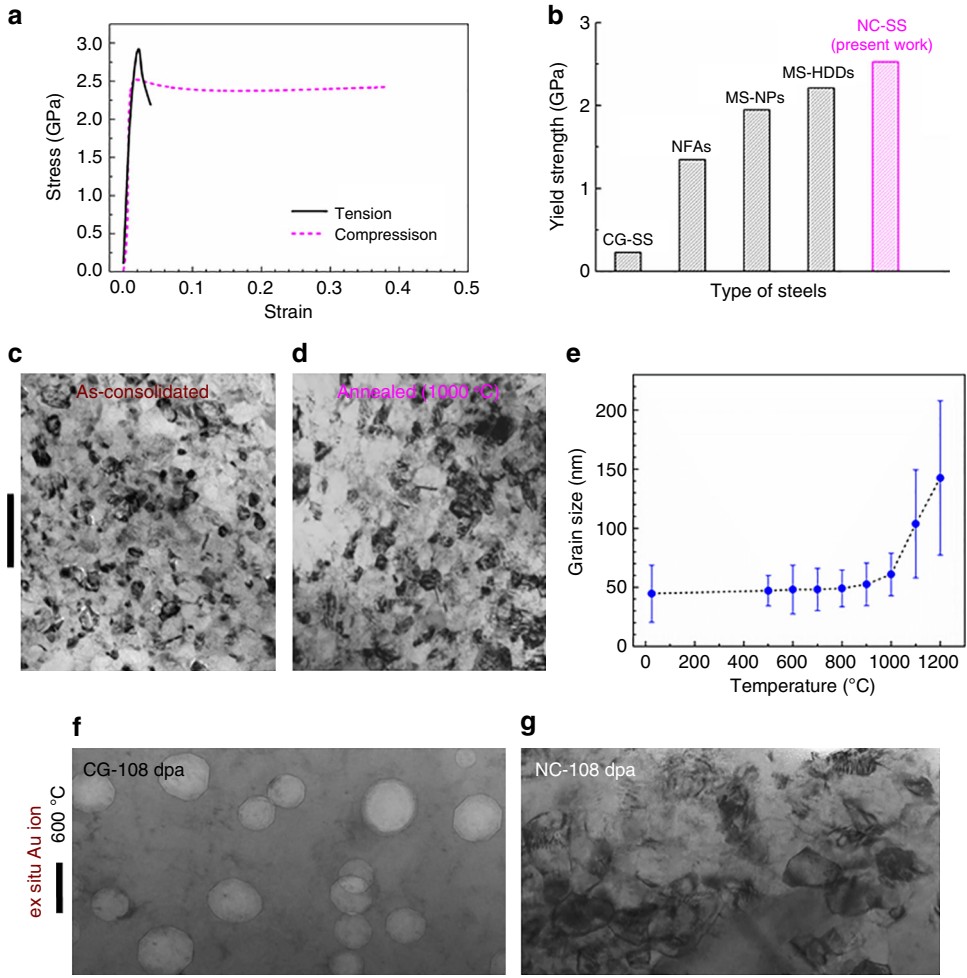

**Fig. 1** Properties and microstructures of NC-SS. **a** True stress–strain curves of NC-SS under tension and compression at a strain rate of $5 \times 10^{-4}$ s$^{-1}$. **b** Yield strength of 304-type CG-SS[2], nanostructured ferritic alloys (NFAs)[44], maraging steel with NPs (MS-NPs)[50], Mn steel with high-density dislocations (MS-HDDs)[51], and present NC-SS. **c**, **d** TEM images of as-consolidated and annealed (at 1000 °C for 1 h) NC-SS. Scale bar, 200 nm. **e** Grain size of as-consolidated NC-SS vs. annealing temperature. **f**, **g** Comparison of the microstructure of CG-SS and NC-SS after ex situ Au ion irradiation to 108 dpa at 600 °C. Both TEM images were chosen between ~200 to 400 nm away from the surface. Scale bar, 50 nm. Irradiation induces a significant number of large voids in CG-SS, while the microstructure of NC-SS remains stable. The average grain size of NC-SS increases slightly from ~45 nm (**c**) to 58 nm (**g**) under intense irradiations

surface of the compressed sample (Supplementary Figure 2a) indeed shows a long crack across the entire sample. In addition, many cracks are observed on the fractured surface of the NC-SS after tensile deformation (Supplementary Figure 2b), suggesting that it is the residual porosity and/or the insufficient particle bonding that causes the low ductility of the NC-SS under tensile deformation.

**Thermal stability**. The NC-SS is composed of nanograins (Fig. 1c) with an average grain size of 45 ± 24 nm. The as-consolidated NC-SS shows an exceptionally high thermal stability against annealing (Fig. 1d, e). No grain growth is observed after annealing at a temperature below 700 °C for 1 h (Fig. 1e). The nanograins grow slightly to ~60 nm after annealing at 1000 °C for 1 h (Fig. 1d). We further annealed the NC-SS at 800 °C for 180 h. The resultant average grain size is 50 ± 15 nm, very close to that of as-consolidated NC-SS.

We also fabricated NC-SS powders without adding elemental La by mechanical milling. The as-milled NC-SS powder has an average grain size of 11 ± 3 nm. However, consolidating this NC-SS powder at 1000 °C under a pressure of 4 GPa yields a SS bulk with an average grain size of ~109 ± 40 nm. Clearly, the La

element plays a critical role in increasing the thermal stability of NC-SS.

**Structure evolution upon irradiation**. The NC-SS exhibits unmatched swelling resistance as compared to conventional CG-SS. Ex situ Au ion irradiation of CG-SS to 108 dpa (peak) at 600 °C leads to a significant number of voids (Fig. 1f), whereas NC-SS irradiated at the same condition has no detectable voids (Fig. 1g). Both TEM images in Fig. 1f, g were chosen between ~200 and 400 nm away from the surface. More details can be found in Supplementary Figure 3. In addition, only slight grain coarsening (from ~45 to 58 nm) is observed in NC-SS after ex situ Au ion irradiation to 108 dpa at 600 °C (Fig. 1g). Note that there is no temperature-induced grain growth when the NC-SS specimen is annealed below 700 °C (Fig. 1e). The slight grain growth for the NC-SS after Au ion irradiation can be ascribed to ion irradiation-induced grain growth. In situ Kr ion irradiation of NC-SS further confirms the high swelling resistance. The BF TEM image and the corresponding selected area diffraction (SAD) patterns (Supplementary Figure 4a) show that 304L CG-SS has a single fcc (γ) phase before irradiation. After in situ Kr ion irradiation to 5 dpa

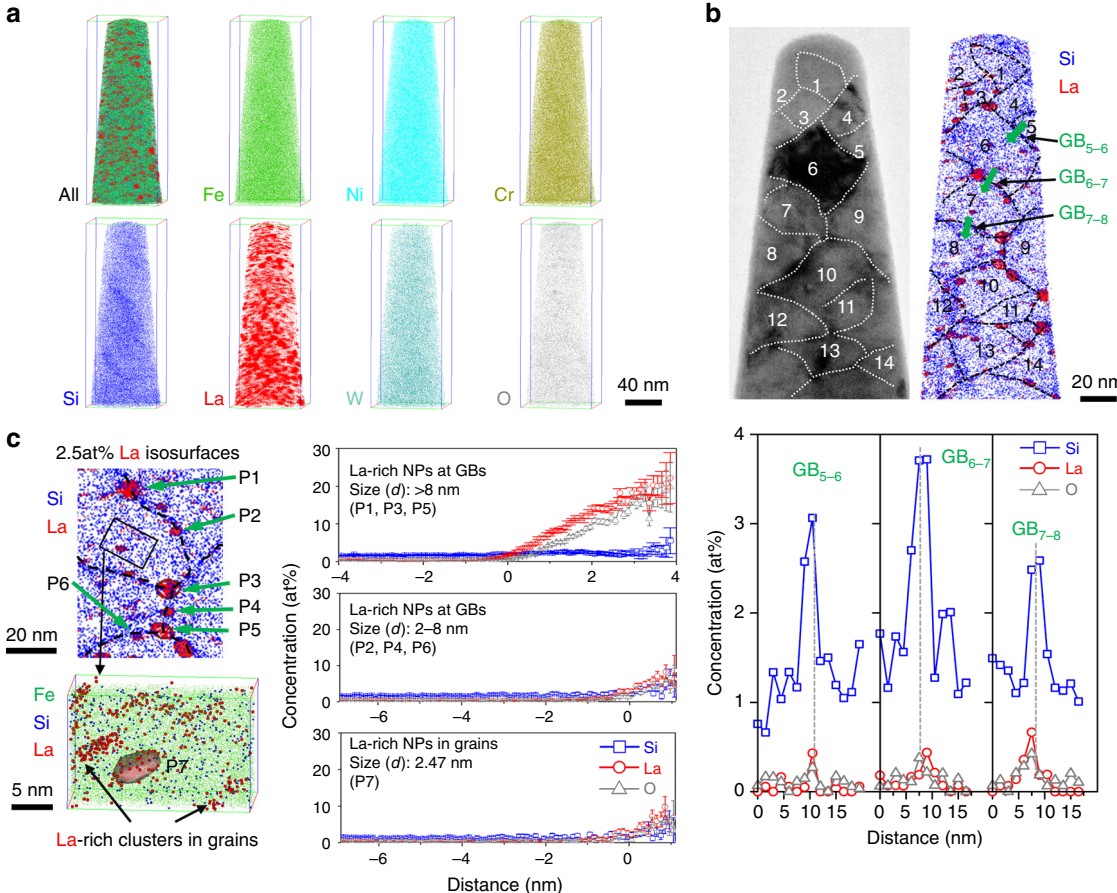

**Fig. 2** Correlative APT and TEM characterization of NC-SS. **a** Combined and individual element atom maps of the analyzed volume. Scale bar, 40 nm. **b** A BF TEM image and corresponding APT Si atom map of a thin slice (5 nm in thickness) reconstructed volume, with 14 resolved nanograins marked as 1–14, respectively and GBs decorated with La-rich NPs. 1D composition profiles across $GB_{5-6}$, $GB_{6-7}$, and $GB_{7-8}$, as marked by green arrows in the Si map, showing the segregation of Si, La, and O at GBs (obtained using an analysis box with a cross-section of $5 \times 5$ nm$^2$, and a bin size of 1.5 nm). Scale bar, 20 nm. **c** A top magnified combined atom map of Si and La from a small reconstructed region with GBs decorated with La-rich NPs, a bottom combined Fe, Si, and La map of a small framed region of a grain containing a La-rich NP defined by iso-surfaces of 2.5 at% La and fine La-rich clusters. The right proxigrams from La-rich NPs in different sizes reveal their compositions of La-rich NPs at GBs and in grain interiors. The particle size of NPs is defined as the equivalent spherical diameter ($d$). Upper scale bar, 20 nm. Lower scale bar, 5 nm

at 500 °C inside a TEM microscope, a significant number of irradiation-induced voids and new phases, including bcc-like (α) and hcp-like (ε) phases, emerge, as shown in Supplementary Figure 4b. In contrast, NC-SS only exhibits moderate phase transformation without detectable voids or grain coarsening after Kr ion irradiation to 40 dpa at 500 °C (Supplementary Figure 4c–f). Comparison of void swelling of neutron and heavy ion irradiated CG-SS, UFG-SS, and NC-SS is summarized in Supplementary Figure 5. The void swelling in this study was measured using the TEM technique[52]. More information is discussed in Supplementary Figure 5. It is worth mentioning that free surfaces of the thin foil may play some roles in defect evolution during in situ irradiation studies. As shown in Supplementary Figure 5, the CG-SS in situ irradiated to 5 dpa at 500 °C has a similar but somewhat higher void swelling as compared to the same type of materials irradiated with neutrons. However, both CG-SS and NC-SS are evaluated under the same in situ irradiation conditions. The thickness of TEM specimens is similar in both cases, ~100 nm. Therefore, the surface effect on radiation-induced defects would be comparable. Noted that NC-SS has an average grain size of ~45 nm, much smaller than the thickness of TEM film. The significantly improved radiation tolerance in NC-SS arises mainly from the high-density GBs, which are dominating sinks for defects.

**Microstructural characterization by TEM and APT**. Abundant NPs are observed (Supplementary Figure 6a) in the NC-SS. These NPs with a size around 5 nm form a semicoherent interface with the matrix (Supplementary Figure 6b). Figure 2a shows the atomic map of different elements in NC-SS. Most elements, except Si, La, and O, appear homogenously distributed in the NC-SS. Correlative transmission electron microscopy (TEM) and atom probe tomography (APT) characterizations, as shown in Fig. 2b, confirm that at least 14 grains, as marked 1–14, are clearly discernable in both the bright field (BF) TEM image and the Si atom map of a thin slice from the analyzed volume of the same APT needle sample. The GBs exhibit slight segregation of Si (Fig. 2b). Typical 1D composition profiles across three GBs were obtained at three corresponding positions marked with green arrows by using an analysis box with a cross-section of $5 \times 5$ nm$^2$ and a bin size of 1.5 nm. The composition profiles reveal significant segregation of Si and slight segregation of La and O at the GBs of the NC-SS. The Gibbs solute excesses of Si, La, and O at the GBs are measured to be $6.42 \pm 0.12$, $0.75 \pm 0.09$, and $0.44 \pm 0.05$ atoms nm$^{-2}$, respectively. Using La iso-surfaces at 2.5 at%, La-rich NPs enriched with Si and O are identified and measured in a high number density of $(5.24 \pm 0.25) \times 10^{23}$ m$^{-3}$ and a high volume fraction of $4.970 \pm 0.003\%$ in the analyzed volume. Chemical analysis shows that nearly 7/9 of the NPs are on the GBs,

and the La-rich NPs on GBs have an average size of $5.2 \pm 2.1$ nm in diameter and a number density of $(4.5 \pm 0.2) \times 10^{23}$ m$^{-3}$. The size distribution of the La(O,Si)-rich NPs in the NC-SS (on the GBs and in the grain interiors) can be found in Supplementary Figure 7. The average size of the NPs is $4.6 \pm 2.4$ nm since the NPs in the grain interiors are smaller than the NPs on the GBs. Only a small fraction of the La-rich NPs is in the grain interiors. A 1D composition profile of a La-rich NP (P7) in the grain interior, as shown in Fig. 2c, reveals that its chemistry is the same as these NPs (P2, P4, P6) with similar sizes on the GBs. Interestingly, careful examinations of Fig. 2c reveal that fine solute clusters enriched with La, O, and Si are frequently present in the grain interiors. Larger La-rich NPs with a size of over 8 nm in diameter are mostly on GBs and highly enriched with La and O, as seen in 1D composition profile (Fig. 2c). The chemical compositions of these La(O,Si)-rich NPs and clusters with different sizes are shown in Supplementary Table 2.

## Discussion

Both the NC-SS and previously studied NFAs have NPs with similar number density and diameter. However, our NC-SS is two to three times stronger than NFAs. Note that the grain size of the NC-SS is approximately an order of magnitude smaller than that of NFAs. Thus, the high yield strength of NC-SS mainly results from GB strengthening rather than dispersion strengthening. The Hall–Petch relation for 304-type CG-SS with grain sizes varying between 1.1 and 50 μm has been established[53]. Extrapolating this relation to a grain size of 45 nm predicts a yield strength of $2.9 \pm 0.2$ GPa, in agreement with our experimentally measured yield strength of $2.5 \pm 0.4$ GPa (Fig. 1a).

Nanograins can be stabilized by both kinetic and thermodynamic strategies. Kinetically, the driving pressure for grain growth due to the curvature of the GB would be counteracted by a pinning (drag) pressure exerted by the particles on the boundary[54]. As a consequence, normal grain growth would be completely inhibited when the grain radius reaches a critical maximum grain radius ($R_C$) given by the Zener equation: $R_C = 4r/3f$, where $R_C$ is the Zener limit, $r$ is the radius of the pinning particles, and $f$ is the volume fraction of particles. Using our experimentally measured $r$ ($2.3 \pm 1.2$ nm) and $f$ (4.97%), one obtains an $R_C$ of $69 \pm 32$ nm. This size is three times as large as the experimentally measured grain radius ($22 \pm 12$ nm, Fig. 1e). This discrepancy suggests that the high thermal stability of our NC-SS can be mainly ascribed to the thermodynamic aspect[6,11–17], i.e., the segregation of elemental La at GBs (Fig. 2b) lowers the specific GB energy, which in turn lowers the thermodynamic driving force for grain growth. Note that an autonomous structural evolution in GBs toward low energy states has been proposed[19] for pure NC Cu and Ni as their grain sizes are reduced below a critical value by plastic deformation. This evolution leads to notable thermal stability in nanograins. This mechanism, however, cannot well explain the high thermal stability achieved in our NC-SS. The grain size of consolidated NC-SS containing 1 at% La is $45 \pm 24$ nm, much smaller than the grain size ($109 \pm 40$ nm) of consolidated NC-SS free from La.

Both GBs and NPs/matrix interfaces are well-known sinks for point defects. The GB area ($A_{GB}$) of a NC material can be estimated as $3/D$, where $D$ is the grain size of the NC material. The NPs/matrix interface area ($A_{int}$) is equal to the product of the number density of NPs and the interface area of each NP. For the present NC-SS, $A_{int}$ ($\approx 3.8 \times 10^7$ m$^{-1}$) is only approximately half $A_{GB}$ ($\approx 6.7 \times 10^7$ m$^{-1}$). These data, together with the fact that the NPs are mainly located on GBs in our NC-SS, suggest that GBs play a major role in suppressing the formation of extended defects. The quantitative point defect capture effectiveness of GBs has been analyzed by using kinetic rate theory[55]. The sink

strength of GBs is given by $S_{GB} = 60/D^2$ when $S^{1/2}D \ll 1$ and by $S_{GB} = 6S^{1/2}/D$ when $S^{1/2}D \gg 1$, where $D$ is the grain diameter and $S$ is the cumulative sink strength of all sinks. Neglecting the point defects captured by the interfaces between NPs and matrix, one obtains $S \approx S_{GB} = 1.8 \times 10^{16}$ m$^{-2}$. At intermediate temperatures, sink strengths greater than $\sim 10^{16}$ m$^{-2}$ are generally needed to provide superior void-swelling resistance in austenitic and ferritic/martensitic steels[56,57]. These high sink strengths are often created by the introduction of a high density of NPs in NFAs. The present work demonstrates a technique to create a high sink strength by ample GBs in a NC-SS.

A microstructure-dependent cluster dynamics model was used to understand the effect of nanoscale grain size and NPs on swelling. The model took the evolution of interstitial and vacancy clusters, defect absorption and emission into account in a NC structure with distributed NPs. Chemical potentials of defects on GBs and NPs' interface were used to describe the interface coherency which determines the defect solubility and sink strength as well as emission efficiency. Migration energies of defects in bulk and on interfaces were used to calculate the inhomogeneous diffusivity of defects. With the model, we simulated the effect of grain sizes and defect chemical potentials at interfaces on defect and defect cluster accumulation. In the simulations, the largest cluster of vacancies and interstitials was set to be 30 defects, respectively. Figure 3a–c shows the distribution of single vacancy concentrations at steady states where the white circles represent the NPs and the white lines denote the GBs. The evolutions of average vacancy and interstitial concentrations for three different microstructures are plotted in Fig. 3d. The results suggest that (i) the smaller the average grain size of the NC-SS, the faster the average defect concentrations reach a steady state; (ii) the average interstitial concentration is much lower than the vacancy concentration because interstitials have a mobility much higher than vacancies and are thus easier to be captured and eliminated by GBs and NP/matrix interfaces; (iii) both average interstitial and vacancy concentrations decrease as the average grain size decreases. Figure 3a–c also shows that the distribution of vacancies is not uniform. Vacancy concentration in a zone near GBs is much lower than that at the center of grains. The thickness of the lower vacancy concentration zone is approx. 6–7 nm, which is almost independent of the average grain sizes considered in the simulations. The highest vacancy concentration at steady state is approx. $3.5 \times 10^{-4}$, as shown in Fig. 3. The evolution of vacancy and interstitial clusters shows that the concentration of clusters with more than 20 defects is zero, and the concentrations of vacancy and interstitial clusters with 2 defects is about $10^{24}$ m$^{-3}$, and with 6 defects is about $10^{12}$ m$^{-3}$. Such a low vacancy and vacancy cluster concentrations indicate that void nucleation may not occur. This agrees well with our experimentally observed results. The defect formation energy depends on the atomistic structure of GBs and NP/matrix interfaces. Chemical potentials of interstitials and vacancies at both the GBs and the NP/matrix interfaces are usually different from those in a perfect crystal. Figure 3e shows the effect of defect chemical potential difference on the evolution of average vacancy and interstitial concentrations for the NC-SS structure with an average grain size of 30 nm. The results show that the average vacancy concentration decreases as the defect chemical potential decreases, i.e., the decrease in the coherency of NP/matrix interface reduces the average vacancy concentration in the matrix. We also simulated the effect of rate constants of clustering, absorption, and emission on defect and defect cluster accumulation. These parameters do not affect the grain size dependence of average vacancy concentration, but affect the absolute value of vacancy and cluster concentrations. A large emission rate of interstitials from NPs cluster reduces the vacancy

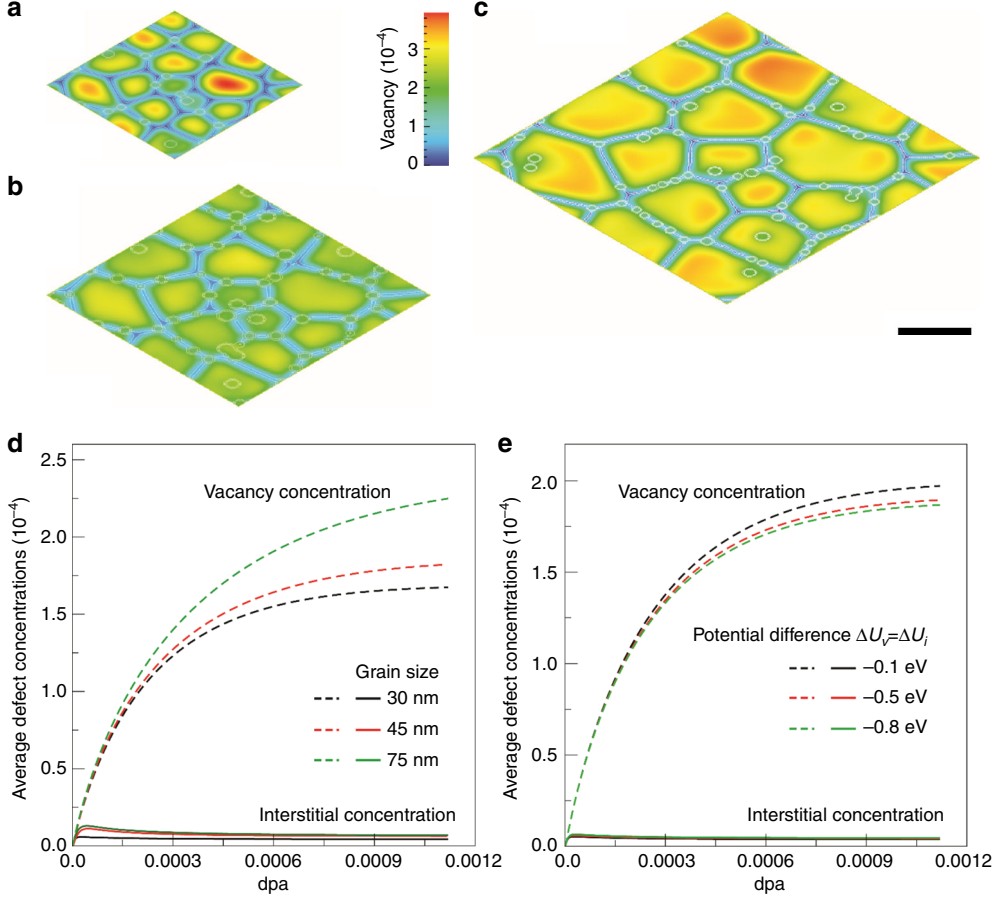

**Fig. 3** Effect of grain sizes and NPs on the vacancy and interstitial concentrations of NC-SS. **a–c** Distributions of vacancy concentrations at steady states for NC-SS structures with an average grain size of 30 nm (**a**), 45 nm (**b**), and 75 nm (**c**). Scale bar, 50 nm. **d** Effect of grain sizes on the evolution of average vacancy and interstitial concentrations. **e** Effect of the difference in defect (vacancy and interstitial) chemical potential $\Delta U$ between NPs' interface and bulk crystal on the evolution of average vacancy and interstitial concentrations. $\Delta U$ varies from −0.1 to −0.8, indicating the decrease of interface coherency. The simulations are performed at 500 °C under a dose rate of $3.0 \times 10^{-4}$ dpa s$^{-1}$

concentration. The results are in agreement with previous simulation results[30,58]. In summary, our simulations confirm that the average vacancy concentration at a steady state decreases as grain size decreases, and reaches $1.8 \times 10^{-4}$ in the NC-SS with an average grain size of 45 nm. NPs in NC-SS can suppress the movement of the GBs, but their effect on radiation resistance depends on the coherency of NP/matrix interfaces, the difference in the mobility between interstitial and vacancy, and the rate constants of absorption and emission.

Rare earth oxide ($Y_2O_3$) is often utilized to form oxide NPs in ODS alloys. The resultant ODS alloys are ultra-fine grained rather than nanocrystalline since these oxide NPs can only kinetically stabilize the grains in ODS alloys. In comparison, rare earth element (La) is added into the present SS alloys. Some elemental La atoms are segregated at GBs and thermodynamically stabilize the nanograins in NC-SS. The remained elemental La atoms form a high density of La-rich NPs mainly distributed along GBs and provide additional kinetic resistance against grain growth. These two factors play a critical role in accomplishing the outstanding thermal stability of nanograins in NC-SS up to 1000 °C, and correspondingly enable unprecedented high mechanical strength, and extraordinary void swelling resistance in austenitic NC-SS at elevated temperatures and high doses. This study provides an important approach to apply nanocrystalline materials for extreme environments. The approach may be applied to other steels and different base metals.

## Methods

**Powder processing and consolidation**. For the preparation of NC-SS, 304-L SS powder (−100 mesh) and lanthanum (La) powder (99.9% pure, −200 mesh), both supplied by Alfa Aesar, were used as starting materials. The powder was weighed to achieve a mixture composed of 99 at% SS and 1 at% La (SS-La) powders. The NC SS-La alloy powder was prepared by mechanically alloying the SS-La powder mixture at room temperature. 8 g of SS and La powders were placed into a tungsten-carbide (WC) vial along with 32 g of WC balls under an Ar atmosphere (containing less than 1 ppm O and $H_2O$) inside a glove box. A SPEX 8000 D shaker mill was used to perform the MA for 24 h. For comparison, NC 304-L SS powder without La addition was also synthesized via the same procedures.

For consolidating the NC powder to bulk, a CS-IB type cubic-anvil apparatus, in which six orthogonal pistons compress a cubic cell volume, was utilized. 6.4 g of NC powder was placed into a cubic-nitride crucible with an inner diameter of 10 mm, an outer diameter of 12 mm, and a height of 15.6 mm. The cubic-nitride crucible was then placed into a graphite crucible with an inner diameter of 12 mm, an outer diameter of 14 mm, and a height of 16.6 mm. The graphite crucible was first placed in the cubic-anvil apparatus and compressed under a pressure of 4 GPa, then heated to 1000 °C at a heating rate of 100 °C min$^{-1}$, and finally remained at 1000 °C under 4 GPa for 30 min. After these procedures, the graphite crucible was cooled to room temperature. Pressure was then fully released to remove the consolidated bulk.

**Alloy composition**. Chemical composition (wt%) of as-received SS powder and consolidated NC SS-La bulk is displayed in Supplementary Table 1. The chemical composition of as-received SS powder is supplied by Alfa Aesar. The chemical composition of consolidated NC SS-La bulk was measured by APT. As-received SS contains such major alloying elements as Cr and Ni and such minor alloying elements as C, Mn, and Si. In addition, as-received SS also contains 0.31 wt% impurity of oxygen. Note that the content of oxygen in as-received SS is very close to that in consolidated SS-La, suggesting that the MA and consolidation processing

do not bring much oxygen impurity. The MA was performed in a tungsten-carbide vial together with tungsten-carbide balls. Thus, approximately 0.5 wt% W impurity was detected in consolidated NC SS-La.

**Annealing.** As-consolidated NC SS-La bulk was annealed in a furnace located in an argon-filled glove-box containing less than 1 ppm O and $H_2O$. Specimens were 1-mm thick disks cut from the as-consolidated bulk. Specimens were heated to the desired annealing temperature (400–1200 °C) at a rate of 100 °C min$^{-1}$, remained at the annealing temperature for 1 h, and then furnace-cooled to room temperature. Specimens were also remained at 800 °C for a time between 1 and 180 h, and then furnace-cooled to room temperature.

**Microstructural characterization.** X-ray diffraction was performed using a RigakuD/MAX/2500/PC X-ray diffractometer with a Cu Kα (λ = 0.154 nm) radiation source. Lattice parameter was determined by extrapolating the Nelson–Riley function[59] $f(\theta) = 0.5 (\cos^2\theta/\sin\theta + \cos^2\theta/\theta)$ to $f(\theta) = 0$, where θ is the diffraction angle. Bright-field TEM was carried out on a JEM-2010 transmission electron microscope operated at 200 kV. The TEM samples were prepared by conventional electropolishing procedures whereby a 3-mm diameter disk was electropolished in an electrolytic solution containing 15 vol% perchloric acid and 85 vol% alcohol under a voltage of 20.5 V. High-resolution TEM (HRTEM) image was conducted in an aberration-corrected environmental TEM, TitanETEMG2, operated at 300 kV. Scanning electron microscopy (SEM) observation was performed in FEI Helios G4 Series DualBeam (Ion/Electron beams) systems operated at a voltage of up to 30 kV.

**Correlative TEM and ATP.** The nanostructure of the NC-SS and distribution of the alloying elements and impurities were investigated by using correlative TEM and APT on the same APT sample. Blanks with a size of $0.5 \times 0.5 \times 15$ mm$^3$ cut from bulk material were electropolished with a standard two-step electropolishing technique to produce APT needle samples. A needle sample was loaded on a specially designed APT sample TEM holder with the maximum tilt angle of ± 70°. TEM examinations were performed by using an FEI Tecnai-T20 TEM at an operation voltage of 200 kV. Subsequent APT characterization of the same needle sample was conducted on a Cameca LEAP 4000X SI instrument, at a specimen base temperature of 40 K, under UV laser pulsing at a pulse laser energy of 40 pJ, a pulse frequency of 250 kHz and a target evaporation rate of 0.5% per pulse. APT data reconstruction and statistical analyses were performed by using a commercial software (Cameca IVAS®3.6.12).

**Mechanical characterization at quasi-static conditions.** The quasi-static compressive test was performed using an Instron 5982 load frame equipped with a 100 kN load cell. The specimen for compression was a cylinder (4 mm in diameter, 6 mm high) made by cutting the as-consolidated NC-SS cylinder (9 mm in diameter, 12 mm high). The quasi-static tensile test was performed using an Instron 5948 load frame equipped with a 2 kN load cell and a non-contact video extensometer. The specimen for tension was made by the following procedures: (i) cutting the as-consolidated NC-SS cylinder into a cuboid ($4.44 \times 4.44$ mm$^2$ in cross-sectional area, 12 mm long), (ii) rolling the cuboid at 900 °C into a plate (1.62 mm thick), and (iii) cutting the plate into a dog-bone shaped specimen ($0.50 \times 0.31$ mm$^2$ in cross-sectional area and 6 mm in gauge length). Both tension and compression tests were conducted at room temperature with a strain rate of $5 \times 10^{-4}$ s$^{-1}$.

**In-situ irradiation.** All TEM specimens were examined by using an FEI Talos 200X microscope before and after irradiation. In situ irradiation experiments were performed at room temperature at the IVEM-TANDEM facility at Argonne National Laboratory. 1 MeV Kr$^{++}$ ion beam was used for irradiation experiments to a maximum fluence of $2.5 \times 10^{16}$ ions cm$^{-2}$ (~40 dpa). The dose rate applied during in situ irradiation experiments was at ~$2.5 \times 10^{-3}$ dpa s$^{-1}$. The Stopping and Range of Ions in Matter (SRIM) (Kinchin–Pease method) simulation was performed to estimate the displacement damage profile (in the unit of displacements-per-atom (DPA)) and Kr ion distribution. Most Kr ions (99.99%) penetrated directly through the TEM specimen and the residual Kr ion concentration in the TEM thin foil is ~0.01 at%. During in situ Kr ion irradiation, the temperature rise of specimens measured by thermocouple was less than 10 °C.

**Ex-situ irradiation.** Ex situ irradiation experiment was performed at room temperature and 600 °C in a $2 \times 1.7$ MV tandem accelerator at Peking University. 6 MeV Au$^{3+}$ ion beam was used for irradiation experiments to a maximum fluence of $1.5 \times 10^{16}$ ions cm$^{-2}$. The dose rate applied during ex situ radiation experiments was kept at ~$2.27 \times 10^{-2}$ ions cm$^{-2}$ s$^{-1}$ for CG-SS and NC-SS. A 2008-SRIM (stopping range of ions in matter) calculation predicts a damage profile extending to 400 nm below the surface, yielding an equivalent peak damage level of 108 dpa (calculated using Kinchin and Pease model with a binding energy of 40 eV), as shown in Supplementary Figure 3. The TEM samples were prepared by mechanical polishing, followed by ion milling to form a wedge to create sufficient electron transparency. TEM observations were carried out with a 200 keV Tecnai F20 microscope at the Electron Microscopy Laboratory of Peking University.

**Microstructure-dependent rate theory simulations.** With the microstructure information of NC-SS including a grain size of 45 nm, a La-riched NPs density of $5.24 \times 10^{23}$ m$^{-3}$, a NP diameter of 5.2 nm, and a spatial distribution of NPs, a phase-field model of multiphase grain growth[60,61] is employed to generate three NC-SS microstructures with an average grain size 30, 45, and 75 nm, respectively. 75% NPs are added on the GBs. Defect absorption and emission take place on GBs and NP/matrix interface. The chemical potentials and mobility of defects at GBs and NP/matrix interfaces are usually different from those inside the grains. In order to describe the inhomogeneous thermodynamic and kinetic properties of defects, two sets of order parameters η and χ are used to describe the grains and NPs, respectively. The order parameters, which are obtained from the phase-field modeling of NC-SS microstructure generation, are 1 inside the grains and NPs, and continuously vary from 1 to 0 across the GBs and NP/matrix interfaces. In this work, we developed a microstructure-dependent cluster dynamics model to investigate the effect of grain sizes and distributed NPs on defect accumulation. Generation, recombination, and clustering of interstitials and vacancies are taken into account in the model. GBs, NPs' interfaces and dislocations are treated as sink and emission sites of defects. Based on the kinetic rate theory and the assumption that only single interstitial and vacancy are mobile, the evolution of defect concentrations can be written as[55,62]:

$$\frac{\partial C_i(x,t)}{\partial t} = \nabla \left[ D_i \nabla C_i(x,t) + D_i C_i(x,t) \nabla \frac{U_i}{k_B T} \right]$$
$$+ G_i - \alpha C_i(x,t) C_j(x,t) + K_j^{li}(2) C_j(x,t) C_{2i}(x,t)$$
$$+ \sum_{m=3}^{M_I} \left[ \gamma_i^{li}(m) - K_i^{li}(m) C_i(x,t) \right] C_{li}(m,x,t)$$
$$- K_i^{li}(2) C_i(x,t) C_{2i}(x,t) - K_i^{li}(1) C_i^2(x,t)$$
$$- \sum_{m=3}^{M_V} \left[ \gamma_i^{lj}(m) - K_j^{lj}(m) C_i(x,t) \right] C_{lj}(m,x,t) - K_i^{lj}(2) C_i(x,t) C_{2j}(x,t)$$
$$+ Z_{i,gb}(\eta)(C_{i_{gb}}^{eq1} - C_i(x,t)) + Z_{i,np}(\chi)(C_{i_{np}}^{eq1} - C_i(x,t))$$
$$+ Z_{i,dis}(C_{i_{dis}}^{eq} - C_i(x,t)) + \xi_{i,gb}(\eta)\left(C_{i_{gb}}^{eq1} - C_i(x,t)\right)$$
$$+ \xi_{i,np}(\chi)\left(C_{i_{np}}^{eq1} - C_i(x,t)\right), i \neq j = \text{Vac and Int}$$

(1)

$$\frac{\partial C_{li}(m,x,t)}{\partial t} = K_i^{li}(m-1) C_i(x,t) C_{li}(m-1,x,t) + \gamma_j^{li}(m-1) C_{li}(m-1,x,t)$$
$$- \{K_i^{li}(m) C_i(x,t) + K_j^{li}(m) C_j(x,t) + \gamma_j^{li}(m)\} C_{li}(m,x,t)$$
$$+ K_i^{li}(m+1) C_j(x,t) C_{li}(m+1,x,t), m = 2,3,4,5,....,M$$
$$i \neq j = \text{Vac and Int}, li = a \text{ cluster with } m \text{ defect } i, M = M_I \text{ or } M_V$$

(2)

where $C_i$ is the concentration of defect $i$; $C_{li}(m)$ is the concentration of defect cluster $li$ consisting of $m$ defects; $D_i$ is the diffusivity of defect $i$; $U_i$ is the interaction energy between sink and defect $i$; $li$ denotes a cluster consisting of defect $i$. $G_i$ denotes vacancy or interstitial generation rate by displacement cascades; α is rate constant for the recombination between single vacancy and interstitials; $K_i^{lj}$ is the rate constant for impingement of defect $i$ to defect cluster $lj$; $\gamma_i^{li}(m)$ is the evolution rate of cluster $m$ of defect $j$ by emitting defect $i$; $Z_{i,def}(\eta, \chi, \rho_{dis})$ is the capture rate of defect $i$ by sinks (def) including grain boundaries (gb), NP interface (np), and dislocation network (dis); and $C_{i_{def}}^{eq1}$ is the equilibrium concentration of defect $i$ on sinks (def); $\xi_{i,def}(\eta, \chi)$ is the emission rate of defect $i$ from sinks (def); $\rho_{dis}$ is the dislocation density; $M_I$ and $M_V$ are the largest sizes of interstitial and vacancy clusters considered, respectively. The thermodynamic and kinetics properties such as $D_i$, $U_i$, α, $K_i^{lj}$, $\gamma_i^{li}(m)$, $Z_{i,def}$, and $C_{i_{def}}^{eq}$ are inhomogeneous and described in terms of the order parameters $\eta_m$ and χ. Diffusivity is calculated by $D_i = D_{0i}\exp(-E_i^m/k_B T)$ where $D_{0i}$ is the diffusion pre-exponential factor and $E_i^m$ is the migration energy of defect $i$; Defect equilibrium concentration $C_{i_{def}}^{eq1}$ is calculated by $\exp(-E_{i_{def}}^f/k_B T)$ where $E_{i_{def}}^f$ is the formation energy of defect $i$ on defect (def); the rate constant $\alpha = 4\pi r_0(D_{int} + D_{vac})$ where $r_0$ is the radius of the recombination volume; rate constant $K_i^{lj} = 4\pi r_{i,j} Z_{lj}^{ij} D_i$ where $r_{i,j}$ is the capture radius between defect $i$ and cluster $lj$; evolution rate $\gamma_i^{lj} = K_i^{lj}/V_{at} \exp(-E_b^i(lj)/k_B T)$ where $V_{at}$ is the atomic volume and $E_b^i(lj)$ is the binding energy between defect $i$ and cluster $lj$. The capture radius $r_{i,j}$ is estimated by $r_{i,j} = (n_{lj})^{1/3} r_{at} + r_{at}$ where $n_{lj}$ is the total number of vacancy/interstitials in cluster $lj$ and $r_{at}$ is the atom radius. The order parameters $\eta_m$ ($m = 1,2,...,m_0$) represent the grain orientations in the polycrystalline structure where $m_0$ is the total number of grains in the simulation cell, and the order parameter χ represents the spatial distribution of NPs. The spatial dependent property $\Phi_i$ is expressed as $\Phi_i = \Phi_{0i} + \Delta\Phi_i f(\eta)$. $\Phi_{0i}$ is the property inside the grains and on the GBs or the NP/matrix interface while $\Delta\Phi_{0i}$ is the difference of the property of defect $i$ at the GB and inside the grain. $f(\eta) = 2.0 \sum_{m=1}^{m_0} (1 - \eta_m)^2$ is a shape function which varies smoothly from 0 inside the grain to 1.0 at the center of GBs. In the simulations, the following parameters: $T = 500$ °C, $r_{at} = 1.41$ Å, $V_{at} = 4\pi/3 r_{at}^3$, $D_{oV} = 4.27 \times 10^{-8}$ m$^2$ s$^{-1}$, $D_{oi} = 2.93 \times 10^{-9}$ m$^2$ s$^{-1}$, $r_0 = 4.5$ Å, $Z_{lj}^i = 1.0$, $Z_{Int,gb}(\eta) = 0.1|\nabla\eta|^2/|\nabla\eta|^2_{\eta=0.5} D_{Int}$, $Z_{Vac,gb}(\eta) = 0.5|\nabla\eta|^2/|\nabla\eta|^2_{\eta=0.5} D_{Vac}$, $Z_{Int,NP}(\chi) = 0.5|\nabla\chi|^2/|\nabla\chi|^2_{\chi=0.5} D_{Int}$,

$Z_{\text{Vac,NP}}(\chi) = 0.5|\nabla\chi|^2/|\nabla\chi|^2_{\chi=0.5}D_{\text{Vac}}$, $\rho_{\text{dis}} = 1 \times 10^8 (m^{-2})$, $Z_{\text{Int,dis}} = 1.1\rho_{\text{dis}}D_{\text{Int}}$, $Z_{\text{Vac,dis}} = 1.0\rho_{\text{dis}}D_{\text{Vac}}$, and $M_I = M_V = 30$ are used. The rest thermodynamic and kinetic properties of austenitic SS are listed in Supplementary Table 3. For defect emission from the sinks, we defined two zones in the simulation cell. One is the emission zone, i.e., the GBs and NP interfaces which is defined by $|\nabla\eta|^2/|\nabla\eta|^2_{\eta=0.5} > 0.02$ and $|\nabla\chi|^2/|\nabla\chi|^2_{\chi=0.5} > 0.02$. The other is the defect accepting zone where the emitted defects can reach during the time increment. The defect accepting zone is a region where any point has the shortest distance to the emission zone that is less than the distance of defect-free path. The distance of defect-free path should be determined by the effective diffusivity of the emitted defect and the time increment, i.e., $R_0 = \sqrt{D^{\text{eff}}\Delta t}$. The effective diffusivity $D^{\text{eff}}$ depends on the density of interacting defects and emitted defect diffusivity. In the simulations, we set it to be one order magnitude higher than that of interstitial diffusivity. The parameter $\xi_{i,\text{gb}}(\eta)$ and $\xi_{i,\text{NP}}(\chi)$ inside the emission zone is set to be 0.6 for interstitials while 0 for vacancies. For every time increment, the total amount of emitted defects from the emission zone is calculated according to emission rate and local defect concentrations described by the last two terms in Eq. (1). The emitted defects are uniformly distributed in the defect accepting zone in current simulations.

## Data availability

The data that support the findings of this study are available from the corresponding author upon reasonable request.

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

## Acknowledgements

This research was supported by the National Natural Science Foundation of China (grant numbers 11575114 and 51571120) and the High-Level Research Program of the Yanshan University (grant number 005000201). S.J. and G.S. thank Dr. J.Z. Liu at the Nanjing University of Science and Technology for his assistance on TEM investigation using a dedicated TEM holder to examine an APT sample. Y.F. and Y.W. were financially supported by the National Magnetic Confinement Fusion Energy Research Project of China (grant number 2015GB113000). S.H. thanks the support from the Pacific Northwest National Laboratory, which is operated by Battelle for the United States Department of Energy under Contract DE-AC05-76RL01830. X.Z. acknowledges financial support from the US NSF-CMMI Program under grant no. 1728419 and acknowledges partial financial support by DOE- Office of Nuclear Energy, NEUP-18-15703. Z.S. was supported by US NSF-DMR-MMN 1611380. The IVEM facility at Argonne National Laboratory is supported by DOE-Office of Nuclear Energy.

## Author contributions

T.S., B.S., and S.X. supervised C.D. and Y.Z. for the synthesis of specimens and the characterization of mechanical property and thermal stability. S.J. and G.S. performed correlative 3D-ATP and TEM experiments and analysis. S.H. carried out simulations. J.H. supervised C.D. and T.Y. for TEM observation and analysis. Y.W. supervised Y.F. for ex-situ irradiation and analysis. X.Z. supervised J.L. and Z.S. for in-situ irradiation studies and analysis. T.S., G.S., J.L., X.Z., and S.H. wrote the paper. T.S. designed and supervised the entire project.

## Additional information

**Competing interests:** The authors declare no competing interests.

