## [Peer Review File · Nature Communications]

Reviewers' comments:

Reviewer #1 (Remarks to the Author):

The manuscript by Du et al. reported a nanocrystalline steel which is thermally stable under annealing up to 1000°C by using lanthanum elements to decorate grain boundaries and to induce oxygen rich precipitates which are believed to stabilize grain boundaries. The steel is tested by using heavy ions such as Kr (in situ) and Au (ex situ) and demonstrates good swelling resistance in comparison with traditional steels.

Grain boundary engineering has been a well-known approach to enhance radiation tolerance. The reported steel, however, represents the most thermally stable steel ever reported so far (to the best of the knowledge of the reviewer). The fact that the grain structures can survive up to 1000 C (or 0.75 Tm) is really a big surprise and this finding alone is publishable. The details given allow repeatable works and statistical analysis is sufficient.

Although exciting, the manuscript has several issues to address.

(1) Phase field modeling is used to compare with experiments. Although the modeling seems to support the experimental observations (that increasing grain boundary density leads to reducing defect concentrations due to boundary defect sink properties), the approach is questionable and is oversimplified. The first issue is that modeling does not include any defect clustering (nucleation and growth of the defect clusters). The irradiation temperatures of 450C and 600C are high enough to induce void and large defect clusters. With or without defect clustering included is a great deal to change the whole picture. Previous studies by Sickfurt and later by Uberuaga (both from Los Alamos Nat. Lab) found that, high density grain boundaries lead to strongly biased interstitial absorption and un-recombined vacancies left behind in domains. Vacancies evolve into bigger clusters and the process leads to final structural collapses. In order to avoid this nanostructure catastrophic failure (which is the opposite of many experimental observations), modeling needs to consider the mechanism that GB can shoot interstitials back to remove supersaturated vacancies behind, which leads to their well cited Science paper (Science 327, 1631 (2010)). The present manuscript does not consider defect clustering (which further forbids GB interstitial shooting back since concentrations of MOBILE interstitial within a domain will never go down without defect-cluster interactions), and it is no surprise that modeling predicts what expected. Although phase field modeling is powerful than MD simulation to predict GB evolutions, the current modeling skips too many detail mechanisms (defect-cluster interactions, defect flux back to cluster rich region, etc..).

(2) The current manuscript explains the mechanism by precipitation stabilization. Recent study reported by Ke Lu group in a Science paper (Enhanced thermal stability of nanograined metals below a critical grain size", Science 360, Issue 6388, pp. 526-530 (2018)) suggests this might be an intrinsic property of nanostructures without any extrinsic impurity/precipitate stabilization mechanisms. Authors need to comments on this finding, which is too important to be ignored.

(3) The effective damage depths by Au ions are quite narrow. The Au implantation peak is only 600 nm. Both injected interstitial effect and surface effects make useful depth region quite narrow. Authors need to specify depth regions for obtaining Fig. 1f and 1g.

(4) In abstract, it is argued that the steel is superior than CG-SS which has a swelling of 1%/dpa. This is a misleading statement. 1%/dpa is the swelling ratio after the swelling incubation period. In principle, all FCC SS expect to swell with 1%/dpa at the very end. Only comparison of incubation period is meaningful.

(5) There are many alloys which show excellent swelling resistance up to 200-400 dpa regions (HT-9, and some ODS alloys). The current alloy was tested to 108 dpa only. Therefore the big

selling point is not superior swelling resistance, but superior GB stability. If swelling is the selling point, much large testing matrix is required.

I recommend a major revision to address the above concerns, prior to its acceptance to publish by Nat. Comm.

Lin Shao of Texas A&M University

Reviewer #2 (Remarks to the Author):

This article provides preliminary information on the strength and radiation resistance of an ultra-strong steel fabricated by novel techniques to produce thermally stable nanocrystalline bulk material. The approach used by the authors appears to be promising for developing new ultra-strong bulk structural alloys. In particular, the thermal stability of the nanocrystalline alloy during short term thermal exposures up to 1000C is significant. However, the use of compressive stress-strain data without any tensile data is a significant limitation. There are numerous examples of high strength materials with good ductility under compressive loads that have very poor tensile ductility; the latter is typically most relevant for structural applications.

The comparison of the yield strength of the present alloy with prior work (Fig. 1b and related text) implies that the present alloy is superior to all other structural materials, which is somewhat misleading. For example, compressive yield strengths of numerous bulk metallic glasses exceed 2.5 GPa, with a maximum reported yield strength more than double that of the present alloy (e.g., Fig. 4 in M. Chen et al., *Annual Reviews Mater. Res.*, vol. 38 (2008) p. 445-469. In the discussion of Fig. 1b, it should also be noted that higher ultimate compressive strengths than the present study have been reported in some steels produced by alternative techniques, e.g., up to 3.5 GPa by S. Dieck, P. Rosemann, A. Kromm and T. Halle, "Reversed austenite for enhancing ductility of martensitic stainless steel", *IOP Conf. Series: Materials Science and Engineering* vol. 181 (2017) 012034.

An alternative approach to produce high strength and radiation resistance is via nanolaminates; for example, strengths of ~2.5 GPa and high radiation tolerance have been reported for such nanolaminates (A. Misra, M.J. Demkowicz, X. Zhang, R.G. Hoagland, *The radiation damage tolerance of ultra-high strength nanolayered composites*, *JOM*, 59 (2007) 62-65). The authors should briefly mention this approach in the manuscript.

It would be valuable to show tensile data to provide information whether there are any flow localization (rapid necking) concerns with this ultra-strong alloy that would not be evident in compression tests. Miniature tensile specimens ~16 mm total length can be used to provide this data on small heats such as used in this study. Without such tensile data the value of the current manuscript is marginal.

(p2-3, lines 89-94) The authors should consider the detection limit for second phases in their XRD system. Unless very long exposures were employed, it is often difficult to detect 1% phases in laboratory XRD systems.

(line 117). the authors should clarify the relevant location of the stated displacement dose for the ion irradiation by changing "108 dpa" to "108 dpa (peak)"

Although advanced characterization was performed on the fabricated alloy, important information on the detailed chemical composition and structure of the La-Si-O nanoparticles in the matrix and along grain boundaries is missing. Are these particles some type of oxide or another structure? On an atomic basis, the powder contained about double the amount of oxygen compared to La.

(lines 109-111). In the discussion of the good thermal stability of the nanocrystals, some mention should be made that longer term annealing studies are needed compared to the 1 h tests in this scoping investigation. Anomalous grain growth phenomena typically emerge at long annealing times and can have devastating impact.

(lines 247-249 and 253-254). The authors claim that incoherent precipitates are not effective as point defect sinks in irradiated materials. This claim is in disagreement with numerous prior experimental (and modeling) studies that have found incoherent precipitate interfaces to be effective in promoting point defect recombination. This calls into question the accuracy of the simulations shown in Fig. 3. The manuscript should be modified accordingly.

The manuscript acknowledgments should list the funding support for the IVEM facility.

There are a few minor typos.

line 79: "much superior to these" should be "be much superior to those".

line 118: "(Fig. 1f). Whereas" should be "(Fig. 1f), whereas".

On p. 3, line 127 "hsown" should be "shown"

On p. 3, lines 132, 133, "Extended Dada" should be "Extended Data"

Reviewer #3 (Remarks to the Author):

The manuscript reports a new nanocrystalline austenitic stainless steel containing 1at.% alloying lanthanum element. The authors claim that the NC-SS has an ultrahigh yield strength of 2.5 GPa, high thermal stability and radiation tolerance. However, the strategy employed in this study to enhance the radiation resistance and thermal stability is not new at all. The experimental methods to evaluate the mechanical property and radiation damage are unfair and cannot support the claim of the research.

By introducing of high density of sinks, such as grain boundaries, interfaces and nanoprecipitates, is an effective way to enhance the radiation tolerance of materials. The current research employed a similar strategy in radiation-resistant materials design, while several important previous works were not cited properly.

The compressive curve of the NC-SS displays an obvious softening after yielding, which means there are damages/cracks developed, indicating the poor deformability of the materials. The authors should show the cross-section image of the compressed sample to demonstrate the damages formed. Actually, it is better to perform tensile test to evaluate the mechanical properties of the NC-SS.

Since the irradiation damage only formed in the top 600 nm, it is not reliable to prove the materials is radiation tolerance by just showing a TEM image (Fig. 1g). Where this sample is cut from? Has it been irradiated by Au ions? The author should show a wide range of the microstructures of the sample from the top surface to the interior and cover the ion irradiated region.

By using high temperature in-situ irradiation test to prove the materials is radiation resistance is also questionable. There is a strong surface effect on the defect dynamics during the thin foils in-situ irradiation, especially at high temperature (500C). Radiation defects (mobile at high temperature) are largely disappeared at top and bottom surfaces, and in this case, what is the effect of grain boundaries? In addition, the low magnification image in extended data figure 2 can hides tiny radiation defects. The authors should show high quality image to demonstrate the microstructures after Kr ion radiation.

For radiation damage only occurs in the range of less than one micrometers (both in-situ and ex-situ radiation), how to measure the void swelling accurately?

There are many typos in the manuscript.

Reviewer #4 (Remarks to the Author):

This paper is about a nanocrystalline austenitic stainless steel, which has been prepared by a powdermetallurgical route and which shows exceptionally high thermal stability, mechanical strength, and irradiation resistance due to a small addition of lanthanum. The presented and discussed results as well as the properties of the designed alloy are indeed exceptional, making the paper worthy of being published at Nature Communications. However, there are some open questions and issues which need to be addressed first, before the paper can be accepted:

p. 2, l. 47-49: "In the kinetic approach, grain boundaries (GBs) are pinned in various ways - such as solute drag, second-phase particle pinning (Zener pinning), chemical ordering, or porosity - to decrease GB mobility."

The authors should give proper references to each of these aspects.

p. 2, l. 51-52: The thermodynamic approach, often achieved by the GB segregation of solutes, has been experimentally studied in NC metals, such as Al₁₁, Cu₁₂, Fe₁₃, and Fe-Cr alloys¹⁴.

Here, the authors do not properly cite the original works of Joerg Weissmueller et al, and Reiner Kirchheim et al, on stabilization of GBs in nanocrystalline alloys by solute segregation and grain boundary energy reduction.

p. 3, l. 92-94: "No diffraction peaks of elemental La are observed in the NC-SS after MA, suggesting that elemental La is incorporated into the lattice of SS matrix by MA."

MA most likely introduces numerous lattice defects such as excess vacancies, dislocations, grain and phase boundaries etc. Thus, La may not be solely solved in the bulk lattice but could also be segregated at lattice defects to a large extent. The authors should consider these possible scenarios in their manuscript as well.

p. 6, l. 175-177: "Typical 1D composition profiles across three GBs were obtained at three corresponding positions marked with green arrows by using an analysis box with a cross-section of 5 × 5 nm² and a bin size of 1.5 nm."

It is not clear why the authors chose a rather small box size and large bin sizes. In my view, using a large box size and smaller bin sizes would make more sense for more efficient atom sampling.

p. 6, l. 179-180: "The Gibbs solute excesses of Si, La and O at the GBs were measured to be 6.42 ± 0.12, 0.75 ± 0.09 and 0.44 ± 0.05 atoms/nm², respectively."

How were the excess values determined? From 1d concentration profiles? It would be more convenient to determine the excess values from ladder diagrams.

p. 6, l. 180 - 182: "Using La iso-surfaces at 2.5 at%, La-rich NPs enriched with Si and O were identified and measured in a high number density of (5.24 ± 0.25) × 10²³ m⁻³ and a high volume fraction of 4.970 ± 0.003% in the analyzed volume."

How was the error for the NP volume fraction determined? It seems unrealistic and too small.

p. 6, l. 212: The term "thermodynamic factor" is used in other contexts and is misleading here. Better use "thermodynamic aspect".

p. 6, l. 214-216: "For the present NC-SS, the interface areas between these NPs and matrix are much less than the GB areas between or among adjacent grains. Thus, GBs play a major role in suppressing the formation of extended defects."

It's hard to understand what the authors mean by the last sentence in this context.

p. 7, l. 223-224: The present work demonstrates a technique to create a high sink strength by GBs in a NC-SS.

Why do GBs in NC-SS show such high sink strength? Is it related to the La NPs?

p. 7, l. 247-248: "NPs with a coherent NP/matrix interface decrease the average vacancy concentration,...?"

Why do coherent NP/matrix interfaces to a decrease in vacancy concentration?

Language revisions:

p. 6, l. 175: "The GBs have slight segregation..." -> "The GBs exhibit slight segregation..."

p. 6, l. 219: "point defect" -> "point defects"

p. 7, l. 257 "again grain growth" -> "against grain growth"

Typos:

p. 3, l. 132 & 133: "Extended dada" -> "Extended data"

p. 6, l. 191: "Lager" -> "Larger"

Reviewer #1 (Remarks to the Author):

The manuscript by Du et al. reported a nanocrystalline steel which is thermally stable under annealing up to 1000°C by using lanthanum elements to decorate grain boundaries and to induce oxygen rich precipitates which are believed to stabilize grain boundaries. The steel is tested by using heavy ions such as Kr (in situ) and Au (ex situ) and demonstrates good swelling resistance in comparison with traditional steels.

Grain boundary engineering has been a well-known approach to enhance radiation tolerance. The reported steel, however, represents the most thermally stable steel ever reported so far (to the best the knowledge of the reviewer). The fact that the grain structures can survive up to 1000 C (or 0.75 Tm) is really a big surprise and this finding alone is publishable. The details given allow repeatable works and statistical analysis is sufficient.

Although exciting, the manuscript has several issues to address.

(1) Phase field modeling is used to compare with experiments. Although the modeling seems to support the experimental observations (that increasing grain boundary density leads to reducing defect concentrations due to boundary defect sink properties), the approach is questionable and is oversimplified. The first issue is that modeling does not include any defect clustering (nucleation and growth of the defect clusters). The irradiation temperatures of 450C and 600C are high enough to induce void and large defect clusters. With or without defect clustering included is a great deal to change the whole picture. Previous studies by Sickfurt and later by Uberuaga (both from Los Alamos Nat. Lab) found that, high density grain boundaries lead to strongly biased interstitial absorption and un-recombined vacancies left behind in domains. Vacancies evolve into bigger clusters and the process leads to final structural collapses. In order to avoid this nanostructure catastrophic failure (which is the opposite of many experimental observations), modeling needs to consider the mechanism that GB can shoot interstitials back to remove supersaturated vacancies behind, which leads to their well cited Science paper (Science 327, 1631 (2010)). The present manuscript does not consider defect clustering (which further forbids GB interstitial shooting back since concentrations of MOBILE interstitial within a domain will never go down without defect-cluster interactions), and it is no surprise that modeling predicts what expected. Although phase field modeling is powerful than MD simulation to predict GB evolutions, the current modeling skips too many detail mechanisms (defect-cluster interactions, defect flux back to cluster rich region, etc.).

Response: Following reviewer's comments, we extended our model to consider the effect of defect clustering. In the new model, chemical potentials of defects on GBs and NPs' interface are used to describe the interface coherency which determines the defect solubility and sink strength as well as emission efficiency. Migration energies of defects in bulk and on interfaces are used to calculate the inhomogeneous diffusivity of defects. For simplicity, we only consider that single vacancy and interstitial are mobile, and the largest cluster of vacancies and interstitials is set to be 30 defects, respectively. With the new model we re-simulated the effect of grain sizes and defect chemical potentials at interfaces on defect and defect cluster accumulation. The new results are presented in the revised manuscript. Compared with previous results, we find that the

- 1) defect clustering does reduce the average vacancy concentration in the matrix, but not much (as shown in Figure 1);
- 2) the concentrations of vacancy and interstitial clusters are similar although vacancy mobility is two order magnitude smaller than that of interstitials (as shown in Figure 2);
- 3) the concentration of clusters with more than twenty defects is zero, and the concentrations of vacancy and interstitial clusters with 2 defects is about $10^{24}(1/m^3)$, and with 6 defects is about $10^{12}(1/m^3)$.

The new results demonstrate a conclusion the same as our previous conclusion, i.e., the average vacancy concentration decreases with the decrease of grain sizes.

(a)

(b)

Figure 1. (a) Previous results without defect clustering, (b) new results with defect clustering

Figure 2. (a) Simulation cell, (b) concentration distributions of interstitial clusters with different interstitials along A-A' line shown in (a), (c) concentration distributions of vacancy clusters with different vacancies along A-A' line shown in (a).

(2) The current manuscript explains the mechanism by precipitation stabilization. Recent study reported by Ke Lu group in a Science paper (Enhanced thermal stability of nanograined metals below a critical grain size”, Science 360, Issue 6388, pp. 526-530 (2018)) suggests this might be an intrinsic property of nanostructures without any extrinsic impurity/precipitate stabilization mechanisms. Authors need to comments on this finding, which is too important to be ignored.

Response: As described in our original manuscript (lines 215 - 231), we explain the extremely high thermal stability by both thermodynamic effect (lowered GB energy by lanthanum segregation at GB) and kinetic effect (numerous nanoprecipitates mainly located at GBs).

We revised the last sentence at the end of the first paragraph in our revised manuscript. The new sentence reads as:
Recent studies show that nanograins can be stabilized by the nanoscale chemical distribution in NC W-Ti alloys²², or by an autonomous grain boundary evolution to low-energy states due to activation of partial dislocations in the plastic deformation of pure NC Cu and Ni²³.

We also added the following discussions to our revised manuscript (lines 225-231):

Note that an autonomous structural evolution in GBs toward low energy states has been proposed²³ for pure NC Cu and Ni as their grain sizes are reduced below a critical value by plastic deformation. This evolution leads to notable thermal stability in nanograins. This mechanism, however, cannot well explain the high thermal stability achieved in our NC-SS. The grain size of consolidated NC-SS containing 1 at% La is 45 ± 24 nm, much smaller than the grain size (109 ± 40 nm) of consolidated NC-SS free from La.

The effective damage depths by Au ions are quite narrow. The Au implantation peak is only 600 nm. Both injected interstitial effect and surface effects make useful depth region quite narrow. Authors need to specify depth regions for obtaining Fig. 1f and 1g.

Response: We appreciate the reviewer's comment. Both TEM images in Fig. 1f and 1g were chosen between ~ 200 nm to 400 nm away from the surface, such information is added in the revised manuscript. We have also modified Extended Data Figure 2 to provide more detailed information.

(4) In abstract, it is argued that the steel is superior than CG-SS which has a swelling of 1%/dpa. This is a misleading statement. 1%/dpa is the swelling ratio after the swelling incubation period. In principle, all FCC SS expect to swell with 1%/dpa at the very end. Only comparison of incubation period is meaningful.

Response: This a very good comment. Little swelling was detected in our NC 304 SS irradiated by both Kr ions (up to 40 dpa) and Au ions (108 dpa). As the reviewer pointed out that the NC 304 SS may still be well within the incubation period, which is surprisingly low compared to a majority of austenitic stainless steels. The manuscript has been revised as "This is in great contrast to the significant void swelling reported for CG-SS to a similar dose level."

(5) There are many alloys which show excellent swelling resistance up to 200-400 dpa regions (HT-9, and some ODS alloys). The current alloy was tested to 108 dpa only. Therefore the big selling point is not superior swelling resistance, but superior GB stability. If swelling is the selling

point, much large testing matrix is required.

Response: We agree with the reviewer on that the superior thermal stability of nanograins is a major selling point. We also agree with the reviewer on the fact that there are many alloys which show excellent swelling resistance up to 200-400 dpa regions (HT-9, and some ODS alloys). However, these alloy are mainly *bcc* (*body-centered-cubic*)-structured ferritic or ferritic/martensitic steels. As we stated in our manuscript, conventional coarse-grained, *face-centered-cubic* (*fcc*) austenitic stainless exhibit poor void swelling resistance in comparison with ferritic steels. In addition, as shown in Extended Data Figure 4, the void swelling of CG-SS after *ex situ* Au ion irradiation to 100 dpa at 600 °C is approx. 9%. In contrast, there is no detectable void swelling in our NC-SS after the same irradiation treatment. Thus, the outstanding void swelling resistance is in great contrast to what has been well known for the poor void swelling tolerance of austenitic stainless steels. The excellent thermal and radiation stability of high angle grain boundaries enables sustainable operation of these defect sinks during high temperature radiations. Consequently our NC 304 SS shows outstanding radiation tolerance, manifested by exceptionally low void swelling after 108 dpa (peak) radiation at elevated temperatures.

I recommend a major revision to address the above concerns, prior to its acceptance to publish by Nat. Comm.

Reviewer #2 (Remarks to the Author):

This article provides preliminary information on the strength and radiation resistance of an ultra-strong steel fabricated by novel techniques to produce thermally stable nanocrystalline bulk material. The approach used by the authors appears to be promising for developing new ultra-strong bulk structural alloys. In particular, the thermal stability of the nanocrystalline alloy during short term thermal exposures up to 1000C is significant. However, the use of compressive stress-strain data without any tensile data is a significant limitation. There are numerous examples of high strength materials with good ductility under compressive loads that have very poor tensile ductility; the latter is typically most relevant for structural applications.

The comparison of the yield strength of the present alloy with prior work (Fig. 1b and related text) implies that the present alloy is superior to all

other structural materials, which is somewhat misleading. For example, compressive yield strengths of numerous bulk metallic glasses exceed 2.5 GPa, with a maximum reported yield strength more than double that of the present alloy (e.g., Fig. 4 in M. Chen et al., Annual Reviews Mater. Res., vol. 38 (2008) p. 445-469. In the discussion of Fig. 1b, it should also be noted that higher ultimate compressive strengths than the present study have been reported in some steels produced by alternative techniques, e.g., up to 3.5 GPa by S. Dieck, P. Rosemann, A. Kromm and T. Halle, “Reversed austenite for enhancing ductility of martensitic stainless steel”, IOP Conf. Series: Materials Science and Engineering vol. 181 (2017) 012034.

Response: The authors tend to agree with the reviewer’s comments. Thus, we have added one sentence in our revised manuscript (lines 114 - 115) to avoid the potential misguidance. The added sentence reads as: These results suggest that the NC-SS reported in this study should be one of the strongest crystalline steels.

The **maximum** compressive strength reported in the publication by S. Dieck et al. (IOP Conf. Series: Materials Science and Engineering vol. 181 (2017) 012034.) is indeed 3.5 GPa. However, the **yield strength** of this alloy is only ~ 1.8 GPa, lower than the **yield strength** of the present nanocrystalline austenitic stainless steel.

An alternative approach to produce high strength and radiation resistance is via nanolaminates; for example, strengths of ~2.5 GPa and high radiation tolerance have been reported for such nanolaminates (A. Misra, M.J. Demkowicz, X. Zhang, R.G. Hoagland, The radiation damage tolerance of ultra-high strength nanolayered composites, JOM, 59 (2007) 62-65). The authors should briefly mention this approach in the manuscript.

Response: This is a good suggestion. The manuscript has been revised and the reference suggested by the reviewer is incorporated in the revised text.

It would be valuable to show tensile data to provide information whether there are any flow localization (rapid necking) concerns with this ultra-strong alloy that would not be evident in compression tests. Miniature tensile specimens ~16 mm total length can be used to provide this data on small heats such as used in this study. Without such tensile data the value of the current manuscript is marginal.

Response: We have done tensile test and revised our manuscript (lines 110 - 118) as below:

Figure 1a shows the compressive and tensile stress-strain curves of consolidated NC-SS bulk. The yield strength in both compression and tension is 2.5 ± 0.4 GPa.These results suggest that the NC-SS should be one of the strongest crystalline steels. Additionally, the NC-SS

under compression exhibits a large fracture strain of ~ 0.4 , indicative of the capability of plastic deformation under compression. In contrast, the NC-SS under tension exhibits rapid necking after its tensile strength reaches 2.9 GPa, leading to a fracture strain of ~ 0.04 .

(p2-3, lines 89-94) The authors should consider the detection limit for second phases in their XRD system. Unless very long exposures were employed, it is often difficult to detect 1% phases in laboratory XRD systems.

Response: The authors agree with the reviewer's comments. We added "within the detection limit of the x-ray diffraction" (line 98). In addition, the following descriptions (lines 101-103) confirm that elemental La is dissolved into the lattice of Fe: The formation of supersaturated solution can be further confirmed by comparing the lattice parameter of NC-SS containing 1at% La, 0.35933 ± 0.00012 nm, with that of NC-SS without La, 0.35827 ± 0.00022 nm.

(line 117). the authors should clarify the relevant location of the stated displacement dose for the ion irradiation by changing "108 dpa" to "108 dpa (peak)"

Response: We have changed "108 dpa" to "108 dpa (peak)". The location of this peak dose is at ~ 400 nm away from the specimen surface, as shown in Extended Data Figure 2.

Although advanced characterization was performed on the fabricated alloy, important information on the detailed chemical composition and structure of the La-Si-O nanoparticles in the matrix and along grain boundaries is missing. Are these particles some type of oxide or another structure? On an atomic basis, the powder contained about double the amount of oxygen compared to La.

Response: Fig.2c proxigrams obtained from La-O-Si-rich nanoparticles in different sizes relative their isosurfaces at 2.5 at% La clearly showed these particles' chemical compositions as well as their neighbour matrix compositions. APT is unable to provide crystallographic information of these particles, rather than their chemistry and size information. It is likely that these fine particles with size < 5 nm are O-La-rich clusters, coherent with the steel matrix, but large ones with sizes over 10 nm may have developed their own lattice structure. In fact, no distinction diffraction information was obtained by TEM analysis, further confirm that they are fine O-La-rich clusters.

(lines 109-111). In the discussion of the good thermal stability of the nanocrystals, some mention should be made that longer term annealing

studies are needed compared to the 1 h tests in this scoping investigation. Anomalous grain growth phenomena typically emerge at long annealing times and can have devastating impact.

Response: The authors agree with the reviewer's comments. We added the following descriptions to our revised manuscript (lines 123 - 124): We further annealed the NC-SS at 800 °C for 180 hours. The resultant average grain size is 50 ± 15 nm, very close to that of as-consolidated NC-SS.

(lines 247-249 and 253-254). The authors claim that incoherent precipitates are not effective as point defect sinks in irradiated materials. This claim is in disagreement with numerous prior experimental (and modeling) studies that have found incoherent precipitate interfaces to be effective in promoting point defect recombination. This calls into question the accuracy of the simulations shown in Fig. 3. The manuscript should be modified accordingly.

Response: We appreciate reviewer's constructive comments. With our modified model, we systematically simulated the effect of interface coherency on average vacancy concentration in the matrix. Our previous model has following simplification: 1) the defect clustering was not considered, 2) emitted interstitial from GBs and NPs' interface were uniformly distributed into the matrix. In the modified model, we considered the defect clustering and defined two zones in the simulation cell for more quantitatively describing interstitial emission. One is the emission zone, i.e., the grain boundaries and NP interfaces which is shown in red in Figure 2(a) (as shown below). The other is the defect accepting zone where the emitted defects can reach during the time increment. The defect accepting zone is a region where any point has a shortest distance to the emission zone that is less than the distance of defect free path, which is shown in green in Figure 2(a). The model, parameters and simulation process are described in the manuscript. We simulated the effect of chemical potentials of defects on NPs' interface, defect free path, rate constants of absorption, and emission on defect at NPs' interface on defect cluster accumulation. The results show that the defect free path is an important parameter which affects the recombination efficiency between vacancies and interstitials in the matrix, hence the average vacancy concentration. In addition, rate constants of absorption, and emission at NPs' interface affect the average vacancy concentration too. As the results reported in reference [47], both the thermodynamic and kinetic properties, which are designing parameters, affect the contribution of NPs on material irradiation resistance. We have more confident conclusions that the average vacancy concentration decreases as the defect chemical potential decreases, i.e., the coherency decrease of NP/matrix interface reduces the average vacancy concentration in the matrix compared to a coherent interface.

For convenience, we copy the Response to Review #1 as below:

Response: We extended our model to consider the effect of defect clustering. In the new model, chemical potentials of defects on GBs and NPs' interface are used to describe the interface coherency which determines the defect solubility and sink strength as well as emission efficiency. Migration energies of defects in bulk and on interfaces are used to calculate the inhomogeneous diffusivity of defects. For simplicity, we only consider that single vacancy and interstitial are mobile, and the largest cluster of vacancies and interstitials is set to be 30 defects, respectively. With the new model we re-simulated the effect of grain sizes and defect chemical potentials at interfaces on defect and defect cluster accumulation. The new results are presented in the revised manuscript. Compared with previous results, we find that the

- 1) defect clustering does reduce the average vacancy concentration in the matrix, but not much (as shown in Figure 1);
- 2) the concentrations of vacancy and interstitial clusters are similar although vacancy mobility is two order magnitude smaller than that of interstitials (as shown in Figure 2);
- 3) the concentration of clusters with more than twenty defects is zero, and the concentrations of vacancy and interstitial clusters with 2 defects is about $10^{24}(1/m^3)$, and with 6 defects is about $10^{12}(1/m^3)$.

The new results demonstrate a conclusion the same as our previous conclusion, i.e., the average vacancy concentration decreases with the decrease of grain sizes.

(a)

(b)

Figure 1. (a) Previous results without defect clustering, (b) new results with defect clustering

Figure 2. (a) Simulation cell, (b) concentration distributions of interstitial clusters with different interstitials along A-A' line shown in (a), (c) concentration distributions of vacancy clusters with different vacancies along A-A' line shown in (a).

The manuscript acknowledgments should list the funding support for the IVEM facility.

Response: We apologize for our overlook. The acknowledgement is added in the manuscript as follow:
 “The IVEM facility at Argonne National Laboratory is supported by DOE-Office of Nuclear Energy.”

There are a few minor typos.

line 79: “much superior to these” should be “be much superior to those”.

line 118: “(Fig. 1f). Whereas” should be “(Fig. 1f), whereas”.

On p. 3, line 127 “hsown” should be “shown”

On p. 3, lines 132, 133, “Extended Dada” should be “Extended Data”

Response: We have corrected these typos.

Reviewer #3 (Remarks to the Author):

The manuscript reports a new nanocrystalline austenitic stainless steel containing 1at.% alloying lanthanum element. The authors claim that the NC-SS has an ultrahigh yield strength of 2.5 GPa, high thermal stability and radiation tolerance. However, the strategy employed in this study to enhance the radiation resistance and thermal stability is not new at all. The experimental methods to evaluate the mechanical property and radiation damage are unfair and cannot support the claim of the research.

Response: Compared with previous studies that often use oxide nanoparticles, e.g., yttrium oxide nanoparticles, as one of the starting materials, we used elemental lanthanum as one of the starting materials. As a result, we stabilized our nanostructured stainless steels to an unprecedented level by using a strategy that combines both thermodynamic (segregation of elemental lanthanum at GBs) and kinetic GB pinning effect (lanthanum-rich oxide nanoprecipitates). The kinetic stabilization approach has been widely used to make ODS steels. However, the resultant grain size of ODS steels is often between 200 and 1,000 nanometers. Our thermodynamic and kinetic stabilization approach is so effective that we could achieve bulk nanocrystalline austenitic stainless steels with an average size below 50 nanometers at a temperature that is 75% of the melting temperature of steels.

Stainless steels are one of the most important nuclear materials. However, their void swelling resistance is very low. As described in the introduction section of our manuscript (lines 78 - 82), “Such GB sinks also operate in austenitic SSs, because decrease in grain size from 50 μm to 450 nm has been reported to be effective in reducing the void swelling⁴¹. Recent results indicate that the void swelling of austenitic UFG-SS with a grain size of 100 nm is nearly an order of magnitude smaller than that of CG-SS⁴².”. The void swelling in our NC-SS with an average grain size below 50 nm is below the detection limit under intense irradiation! Furthermore the outstanding thermal stability of these nanograins ensures that GBs remain as active defect sinks at high temperature, whereas most conventional nanostructured steels would coarsen significantly during radiation.

Both compression and tension tests show that the yield strength of NC-SS is ~ 2.5 GPa as shown in the revised Fig. 1a.

By introducing of high density of sinks, such as grain boundaries, interfaces and nanoprecipitates, is an effective way to enhance the radiation tolerance of materials. The current research employed a similar strategy in radiation-resistant materials design, while several important previous works were not cited properly.

Response: Grain boundaries are one of the most intensively investigated defect sinks for radiation studies. The stability of grain boundaries in irradiated nanocrystalline metals is a major concern [X. Zhang et al. / Progress in Materials Science 96 (2018) 217.]. Hence the current study provides a convincing example where the radiation and thermal stability of grain boundaries can be significantly improved.

We added the following descriptions (lines 67 - 69) to the revised manuscript:

Nanostructured materials, such as nanotwinned metals^{32,33}, nanolaminates^{34,35}, nanocrystalline metals^{33,36}, etc., with unique predesigned defect sinks have the potential to provide both high strength and radiation resistance.

The compressive curve of the NC-SS displays an obvious softening after yielding, which means there are damages/cracks developed, indicating the poor deformability of the materials. The authors should show the cross-section image of the compressed sample to demonstrate the damages formed. Actually, it is better to perform tensile test to evaluate the mechanical properties of the NC-SS.

Response: we have added tensile test to the revised manuscript. Both compression and tension tests show that the yield strength of NC-SS is ~ 2.5 GPa as shown in the revised Fig. 1a.

In addition, we revised our manuscript (lines 110 - 118) as below:

Figure 1a shows the compressive and tensile stress-strain curves of consolidated NC-SS bulk. The yield strength in both compression and tension is 2.5 ± 0.4 GPa. These results suggest that the NC-SS should be one of the strongest crystalline steels. Additionally, the NC-SS under compression exhibits a large fracture strain of ~ 0.4, indicative of the capability of plastic deformation under compression. In contrast, the NC-SS under tension exhibits rapid necking after its tensile strength reaches 2.9 GPa, leading to a fracture strain of ~ 0.04.

Since the irradiation damage only formed in the top 600 nm, it is not reliable to prove the materials is radiation tolerance by just showing a TEM

image (Fig. 1g). Where this sample is cut from? Has it been irradiated by Au ions? The author should show a wide range of the microstructures of the sample from the top surface to the interior and cover the ion irradiated region.

Response: We appreciate the reviewers comment. Both TEM images in Fig. 1f and 1g were chosen between ~ 200 nm to 400 nm away from the surface, such information is added in the revised manuscript. We have also revised Extended Data Figure 2 to show a wide range of the microstructures of the sample from the top surface to the interior and cover the ion irradiated region.

By using high temperature in-situ irradiation test to prove the materials is radiation resistance is also questionable. There is a strong surface effect on the defect dynamics during the thin foils in-situ irradiation, especially at high temperature (500C). Radiation defects (mobile at high temperature) are largely disappeared at top and bottom surfaces, and in this case, what is the effect of grain boundaries? In addition, the low magnification image in extended data figure 2 can hides tiny radiation defects. The authors should show high quality image to demonstrate the microstructures after Kr ion radiation.

Response: We thank the reviewers for good comments and would like to answer the questions from the following aspects:

(i) In this study, we have used both *ex situ* and *in situ* irradiation techniques. Both studies showed that NC 304 SS exhibited significantly improved irradiation resistance, manifested by much less swelling comparing to coarse-grained counterparts. Hence the results of in situ and ex situ radiation are consistent with each other.

(ii) *In situ* heavy ion irradiation has been widely used to evaluate irradiation resistance of various materials [X. Zhang et al., Progress in Materials Science 96 (2018) 217; M. Li et al., Philosophical Magazine 92 (2012) 2048; M. Li et al., Journal of Nuclear Materials 498 (2018) 199; M. L. Jenkins and M. A. Kirk, Characterization of Radiation Damage by Transmission Electron Microscopy]. We agree that free surfaces of the thin foil play some roles in defect evolution. As shown in Extended Data Figure 4 in the current study, the CG 304 SS in situ irradiated to 5 dpa at 500 °C (black triangle) has similar but somewhat higher void swelling as compared to the same type of materials irradiated with neutrons. Hence we anticipate that the in situ Kr ion irradiation studies produce void swelling results that are somewhat comparable to neutron radiation induced swelling in 304 SS.

Extended Data Figure 4. Comparison of void swelling of neutron and heavy ion irradiated CG-SS (Refs. 57, 58 and this study), UFG-SS (Ref. 57) and NC-SS (this study) at a temperature between 400 and 600 °C. All SS are 304L type. Void swelling cannot be detected in NC-SS after both *in situ* Kr ion irradiation (40 dpa, 500°C) and *ex situ* Au ion irradiation (108 dpa, 600 °C).

In the current study, we evaluated both CG 304 SS and NC 304 SS under the same *in situ* irradiation conditions (temperature). The thickness of TEM specimens is similar in both cases, ~ 100 nm. Hence the surface effect on radiation induced defects would be comparable. NC 304 SS has an average grain size of ~ 45 nm, much smaller than that TEM film thickness. The significantly improved radiation tolerance in NC 304 SS arises mainly from the high-density grain boundaries, which are dominating sinks for defects.

(iii) A higher resolution and high magnification TEM image (shown below) shows the microstructure of NC 304 SS after irradiation to 40 dpa at 500 °C. Voids were not detected.

TEM image of NC 304 SS after in situ irradiation at 500 °C to 40 dpa.

For radiation damage only occurs in the range of less than one micrometers (both in-situ and ex-situ radiation), how to measure the void swelling accurately?

Response: This is a very good question. In general two methods have been used to probe void swelling in heavy ion irradiated specimens. One of them is to measure the step height variation by using a mask, as described in [E.G. Fu et al, Journal of Nuclear Materials 407 (2010) 178]. Another method used in this study is to measure void size and density from TEM studies and then calculate void swelling.

There are many typos in the manuscript.

Response: We have corrected these typos.

Reviewer #4 (Remarks to the Author):

This paper is about a nanocrystalline austenitic stainless steel, which has been prepared by a powder metallurgical route and which shows exceptionally high thermal stability, mechanical strength, and irradiation resistance due to a small addition of lanthanum. The presented and discussed results as well as the properties of the designed alloy are indeed exceptional, making the paper worthy of being published at Nature Communications. However, there are some open questions and issues which need to be addressed first, before the paper can be accepted:

p. 2, l. 47-49: “In the kinetic approach, grain boundaries (GBs) are pinned in various ways - such as solute drag, second-phase particle pinning (Zener pinning), chemical ordering, or porosity - to decrease GB mobility.”

The authors should give proper references to each of these aspects.

Response: We have added proper references to each of these aspects.

p. 2, l. 51-52: The thermodynamic approach, often achieved by the GB segregation of solutes, has been experimentally studied in NC metals,

such as Al₁₁, Cu₁₂, Fe₁₃, and Fe-Cr alloys¹⁴.

Here, the authors do not properly cite the original works of Joerg Weissmueller et al, and Reiner Kirchheim et al, on stabilization of GBs in nanocrystalline alloys by solute segregation and grain boundary energy reduction.

Response: We have cited the original works of Joerg Weissmueller and Reiner Kirchheim on the stabilization of GBs in nanocrystalline alloys by solute segregation. There references are:

Weissmüller J. Alloy effects in nanostructures. *Nanostruct.Mater.* **3**, 261-272 (1993).

Weissmüller J. Alloy thermodynamics in nanostructures. *J. Mater. Res.* **9**, 4-7 (1994).

Kirchheim R. Grain coarsening inhibited by solute segregation. *Acta Mater.* **50**, 413-419 (2002).

p. 3, l. 92-94: “No diffraction peaks of elemental La are observed in the NC-SS after MA, suggesting that elemental La is incorporated into the lattice of SS matrix by MA.”

MA most likely introduces numerous lattice defects such as excess vacancies, dislocations, grain and phase boundaries etc. Thus, La may not be solely solved in the bulk lattice but could also be segregated at lattice defects to a large extent. The authors should consider these possible scenarios in their manuscript as well.

Response: The authors agree with the reviewer’s comments. We added the following descriptions to the revised manuscript (line 103 - 106): In addition, note that MA technique often introduces numerous lattice defects such as excess vacancies, dislocations, and grain boundaries etc. Thus, La may not be solely dissolved in the bulk lattice but could also be segregated at these lattice defects to a large extent.

p. 6, l. 175-177: “Typical 1D composition profiles across three GBs were obtained at three corresponding positions marked with green arrows by using an analysis box with a cross-section of $5 \times 5 \text{ nm}^2$ and a bin size of 1.5 nm.”

It is not clear why the authors chose a rather small box size and large bin sizes. In my view, using a large box size and smaller bin sizes would make more sense for more efficient atom sampling.

Response: Since the GBs were highly decorated by fine O-La-rich clusters, their spacing is very small as seen in Fig 2c. We used an analysis box with such a small cross section to make sure no small GB clusters were included in the measurement in order to get consistent pure segregation information of these GBs. To get better statistics, we use a bin size of 1.5 nm in order to include sufficient number of atoms (at least 600 atoms/each bin) in our analysis to avoid to high uncertainty for our measurement results. Indeed, if there were no such limitation, we would often use a large box size and a small bin in our analysis for GB segregation.

p. 6, l. 179-180: “The Gibbs solute excesses of Si, La and O at the GBs were measured to be 6.42 ± 0.12 , 0.75 ± 0.09 and 0.44 ± 0.05 atoms/nm², respectively.”

How were the excess values determined? From 1d concentration profiles? It would be more convenient to determine the excess values from ladder diagrams.

Response: Yes, it was determined by using 1D concentration profiles. In fact, we also checked by using ladder diagrams, both gave the similar values. As both methods in solute excess measurement have been widely reported in literature, we list the values from the 1D concentration profiles.

p. 6, l. 180 - 182: “Using La iso-surfaces at 2.5 at%, La-rich NPs enriched with Si and O were identified and measured in a high number density of $(5.24 \pm 0.25) \times 10^{23} \text{ m}^{-3}$ and a high volume fraction of $4.970 \pm 0.003\%$ in the analyzed volume.”

How was the error for the NP volume fraction determined? It seems unrealistic and too small.

Response: A high number (more than 500 actually) of La-O-Si-rich particles have been identified in analysis volume of APT using the iso-surfaces at 2.5 at%-La. The error of the volume fraction is given by square root of the total atoms (~ 2.7 million) in the O-La-Si-rich clusters

over the total atoms (~ 65 million) in the analyzed volume, which is very small indeed.

p. 6, l. 212: The term “thermodynamic factor” is used in other contexts and is misleading here. Better use “thermodynamic aspect”.

Response: We have replaced “thermodynamic factor” by “thermodynamic aspect”.

p. 6, l. 214-216: “For the present NC-SS, the interface areas between these NPs and matrix are much less than the GB areas between or among adjacent grains. Thus, GBs play a major role in suppressing the formation of extended defects.”

It’s hard to understand what the authors mean by the last sentence in this context.

Response: We thank the reviewer for this good comment. We have replaced these two sentences by the following descriptions (lines 232 - 237): Both GBs and NPs/matrix interfaces are well-known sinks for point defects. The GB area (A_{GB}) of a NC material can be estimated as $3/D$, where D is the grain size of the NC material. The NPs/matrix interface area (A_{int}) is equal to product of the number density of NPs and the interface area of each NP. For the present NC-SS, A_{int} ($\approx 3.8 \times 10^7 \text{ m}^{-1}$) is only ~ half A_{GB} ($\approx 6.7 \times 10^7 \text{ m}^{-1}$). These data, together with the fact that the NPs are mainly located on GBs in our NC-SS, suggest that GBs play a major role in suppressing the formation of extended defects.

p. 7, l. 223-224: The present work demonstrates a technique to create a high sink strength by GBs in a NC-SS.

Why do GBs in NC-SS show such high sink strength? Is it related to the La NPs?

Response: As we described in our manuscript (lines 239 - 242): The sink strength of GBs is given by $S_{GB} = 60/D^2$ when $S^{1/2}D \ll 1$ and by $S_{GB} = 6S^{1/2}/D$ when $S^{1/2}D \gg 1$, where D is the grain diameter and S is the cumulative sink strength of all sinks. Neglecting the point defect captured by the interfaces between NPs and matrix, one obtains $S \approx S_{GB} = 1.8 \times 10^{16} \text{ m}^{-2}$.

For the present NC-SS, $S^{1/2}D \gg 1$. Thus, neglecting the point defect captured by the interfaces between NPs, one obtains $S \approx S_{GB} = 36/D^2 = 1.8 \times 10^{16} \text{ m}^{-2}$. This sink strength is estimated by only considering the GBs. GBs in any NC material should provide a high sink strength due to its small grain size D .

p. 7, l. 247-248: “NPs with a coherent NP/matrix interface decrease the average vacancy concentration,...?
Why do coherent NP/matrix interfaces to a decrease in vacancy concentration?”

Response: The distance of defect free path is an important parameter which affects the efficiency of defect recombination. For convenience we copy the response to other reviewers as below:

Response to Reviewer #1:

We extended our model to consider the effect of defect clustering. In the new model, chemical potentials of defects on GBs and NPs' interface are used to describe the interface coherency which determines the defect solubility and sink strength as well as emission efficiency. Migration energies of defects in bulk and on interfaces are used to calculate the inhomogeneous diffusivity of defects. For simplicity, we only consider that single vacancy and interstitial are mobile, and the largest cluster of vacancies and interstitials is set to be 30 defects, respectively. With the new model we re-simulated the effect of grain sizes and defect chemical potentials at interfaces on defect and defect cluster accumulation. The new results are presented in the revised manuscript. Compared with previous results, we find that the

- 1) defect clustering does reduce the average vacancy concentration in the matrix, but not much (as shown in Figure 1);
- 2) the concentrations of vacancy and interstitial clusters are similar although vacancy mobility is two order magnitude smaller than that of interstitials (as shown in Figure 2);
- 3) the concentration of clusters with more than twenty defects is zero, and the concentrations of vacancy and interstitial clusters with 2 defects is about $10^{24} (1/\text{m}^3)$, and with 6 defects is about $10^{12} (1/\text{m}^3)$.

The new results demonstrate a conclusion the same as our previous conclusion, i.e., the average vacancy concentration decreases with the decrease of grain sizes.

(a)

(b)

Figure 1. (a) Previous results without defect clustering, (b) new results with defect clustering.

Figure 2. (a) Simulation cell, (b) concentration distributions of interstitial clusters with different interstitials along A-A' line shown in (a), (c) concentration distributions of vacancy clusters with different vacancies along A-A' line shown in (a).

Response to Reviewer #2:

With our modified model, we systematically simulated the effect of interface coherency on average vacancy concentration in the matrix. Our previous model has following simplification: 1) the defect clustering was not considered, 2) emitted interstitial from GBs and NPs' interface were uniformly distributed into the matrix. In the modified model, we considered the defect clustering and defined two zones in the simulation cell for more quantitatively describing interstitial emission. One is the emission zone, i.e., the grain boundaries and NP interfaces which is shown in red in Figure 2(a) (as shown above). The other is the defect accepting zone where the emitted defects can reach during the time increment. The defect accepting zone is a region where any point has a shortest distance to the emission zone that is less than the distance of defect free path, which is shown in green in Figure 2(a). The model, parameters and simulation process are described in the manuscript. We simulated the effect

of chemical potentials of defects on NPs' interface, defect free path, rate constants of absorption, and emission on defect at NPs' interface on defect cluster accumulation. The results show that the defect free path is an important parameter which affects the recombination efficiency between vacancies and interstitials in the matrix, hence the average vacancy concentration. In addition, rate constants of absorption, and emission at NPs' interface affect the average vacancy concentration too. As the results reported in reference [47], both the thermodynamic and kinetic properties, which are designing parameters, affect the contribution of NPs on material irradiation resistance. We have more confident conclusions that the average vacancy concentration decreases as the defect chemical potential decreases, i.e., the coherency decrease of NP/matrix interface reduces the average vacancy concentration in the matrix compared to a coherent interface.

Language revisions:

p. 6, l. 175: "The GBs have slight segregation..." -> "The GBs exhibit slight segregation..."

p. 6, l. 219: "point defect" -> "point defects"

p. 7, l. 257 "again grain growth" -> "against grain growth"

Response: We thank the reviewer for careful reading. We have completed these language revisions.

Typos:

p. 3, l. 132 & 133: "Extended dada" -> "Extended data"

p. 6, l. 191: "Lager" -> "Larger"

Response: We thank the reviewer for careful reading. We have corrected these typos.

Reviewers' comments:

Reviewer #1 (Remarks to the Author):

The overall response is satisfactory. I appreciated the efforts to re-simulate defect interactions by including defect clustering into the picture. The manuscript is recommended to publish as it is. One reviewer pointed out that the approach is not innovative enough. It is true that there are so many previous reports already on boundary engineering. But the approach reported here is more realistic than others for industry scale up process and the finding on its superior stability under extreme high temperature is quite important to the field. Hence, its impact is appropriate for Nat. Comm.

Reviewer #2 (Remarks to the Author):

The changes made by the authors in response to the referee comments are satisfactory.

Reviewer #3 (Remarks to the Author):

The authors addressed some of the concerns raised during first review, but some important issues were not touched. The remained questions and concerns are listed below again.

The advantage of elemental lanthanum need to be explained further in the manuscript.

By introducing of high density of sinks, such as grain boundaries, interfaces and nanoprecipitates, is an effective way to enhance the radiation tolerance of materials. The current research employed a similar strategy in radiation-resistant materials design, while several important previous works were not cited properly, especially those related to sink efficiency of GBs and interfaces.

The compressive curve of the NC-SS displays an obvious softening after yielding, which means there are damages/cracks developed, indicating the poor deformability of the materials. The authors should show the cross-section image or morphology of the compressed sample to demonstrate the damages formed.

The tensile ductility of the NC-SS is quite small, only a few percent. Is such a brittle metal applicable in real industry?

The strong surface effect of TEM thin sample on the defect dynamics during in-situ irradiation need to be addressed and discussed, especially at high temperature (500C).

The error bar for the void swelling measurement need to be discussed and marked. The relationship and difference between the current methods and the bulk measurement need to be discussed as well.

Reviewer #4 (Remarks to the Author):

The authors have addressed all the questions and comments I raised. In my view, the manuscript can be accepted for publication at Nature Communications.

Reviewers' comments:

Reviewer #1 (Remarks to the Author):

The overall response is satisfactory. I appreciated the efforts to re-simulate defect interactions by including defect clustering into the picture. The manuscript is recommended to publish as it is. One reviewer pointed out that the approach is not innovative enough. It is true that there are so many previous reports already on boundary engineering. But the approach reported here is more realistic than others for industry scale up process and the finding on its superior stability under extreme high temperature is quite important to the field. Hence, its impact is appropriate for Nat. Comm.

Reviewer #2 (Remarks to the Author):

The changes made by the authors in response to the referee comments are satisfactory.

Reviewer #3 (Remarks to the Author):

The authors addressed some of the concerns raised during first review, but some important issues were not touched. The remained questions and concerns are listed below again.

The advantage of elemental lanthanum need to be explained further in the manuscript.

Response: we revised the last paragraph (lines 302-308) to emphasize the advantage of elemental lanthanum. The revised paragraph reads as:

Rare earth oxide (Y_2O_3) is often utilized to form oxide NPs in ODS alloys. The resultant ODS alloys are ultra-fine grained rather than nanocrystalline since these oxide NPs can only kinetically stabilize the grains in ODS alloys. In comparison, rare earth element (La) is added into the present SS alloys. Some elemental La atoms are segregated at GBs and thermodynamically stabilize the nanograins in NC-SS. The remained elemental La atoms form a high density of La-rich NPs mainly distributed along GBs and provide additional kinetic resistance against grain growth.

By introducing of high density of sinks, such as grain boundaries, interfaces and nanoprecipitates, is an effective way to enhance the radiation tolerance of materials. The current research employed a similar strategy in radiation-resistant materials design, while several important previous works were not cited properly, especially those related to sink efficiency of GBs and interfaces.

Response: we have added one more sentence (lines 67-68) to address this issue. The sentence reads as:

Introducing a high density of sinks, such as grain boundaries³²⁻³⁵, interfaces³⁶⁻⁴⁰ and nanoprecipitates^{41,42}, is an effective way to enhance the radiation tolerance of materials.

References:

32. Bai, X. M., Voter, A. F., Hoagland, R. G., Nastasi, M. & Uberuaga, B. P. Efficient annealing of radiation damage near grain boundaries via interstitial emission. *Science* **327**, 1631-1634 (2010).

33. Han, W. Z., Demkowicz, M. J., Fu, E. G., Wang, Y. Q., Misra A. Effect of grain boundary character on sink efficiency. *Acta Mater.* **60**, 6341-6351 (2012)
34. Tschopp, M. A., Solanki, K. N., Gao, F., Sun, X., Khaleel, M. A., Horstemeyer, M. F. Probing grain boundary sink strength at the nanoscale: Energetics and length scales of vacancy and interstitial absorption by grain boundaries in α -Fe. *Phys. Rev. B* **85**, 064108 (2012).
35. Zhang, X., Hattar, K., Chen, Y., Shao, L., LI, J., Sun, C., Yu, K., LI, N., Taheri, M. L., Wang, H., Wang, J., Nastasi, M. Radiation damage in nanostructured materials. *Prog. Mater. Sci.* **96**, 217-321 (2018).
36. Misra, A., Demkowicz, M. J., Zhang X., Hoagland, R. G. The radiation damage tolerance of ultra-high strength nanolayered composites. *JOM* **59**, 62-65 (2007).
37. Demkowicz, M. J., Hoagland, R. G., Hirth, J. P. Interface structure and radiation damage resistance in Cu-Nb multilayer nanocomposites. *Phys. Rev. Lett.* **100**, 136102 (2008).
38. Beyerlein, I. J., Caro, A., Demkowicz, M. J., Mara, N. A., Misra, A., Uberuaga, B. P. Radiation damage tolerant nanomaterials. *Mater. Today* **16**, 443-449 (2013).
39. Han, W., Demkowicz, M. J., Mara, N. A., Fu, E., Sinha, S., Rollett, A. D., Wang, Y., Carpenter, J. S., Beyerlein, I. J., Misra, A. Design of radiation tolerant materials via interface engineering. *Adv. Mater.* **25**, 6975-6979 (2013).
40. Beyerlein, I. J., Demkowicz, M. J., Misra, A., Uberuaga, B. P. Defect-interface interactions. *Prog. Mater. Sci.* **74**, 125-210 (2015).
41. Zinkle, S. J., Busby, J. T. Structural materials for fission & fusion energy. *Mater. Today* **12**, 12-19 (2009).
42. Hsiung, L. L., Fluss, M. J., Tumey, S. T., Choi, B. W., Serruys, Y., Willaime, F., Kimura, A. Formation mechanism and the role of nanoparticles in Fe-Cr ODS steels developed for radiation tolerance. *Phys. Rev. B* **82**, 184103 (2010).

The compressive curve of the NC-SS displays an obvious softening after yielding, which means there are damages/cracks developed, indicating the poor deformability of the materials. The authors should show the cross-section image or morphology of the compressed sample to demonstrate the damages formed.

Response: In order to address this issue, we have added a couple of sentences (lines 120-126) and SEM images [Extended Data Figure 2. SEM images of the top surface of NC-SS cylinder after compressive deformation (a) and the fractured surface of NC-SS after tensile deformation (b).]. The added text reads as:

Note that the compressive curve of the NC-SS displays a softening after yielding, suggesting that there are damages and/or cracks developed. The morphology of the top surface of the compressed sample (Extended Data Figure 2a) indeed shows a long crack across the whole sample. In addition, many cracks are observed on the fractured surface of the NC-SS after tensile deformation (Extended Data Figure 2b), suggesting that it is the residual porosity and/or the insufficient particle bonding that causes the low ductility of the NC-SS under tensile deformation.

The tensile ductility of the NC-SS is quite small, only a few percent. Is such a brittle metal

applicable in real industry?

Response: Our research team takes a step-wise approach to solve the current challenge of design of advanced radiation tolerant materials. It is difficult to identify a solution that can solve all challenging issues at the same time. Although the current nanocrystalline 304 SS has outstanding thermal stability, radiation tolerance and high compressive strength and plasticity, there is still a long way before the implement of such materials in real nuclear reactor environments. One of the drawbacks in the current NC 304SS is limited ductility as pointed out by the reviewers. As a matter of fact, it has been well known for decades that the tensile ductility of NC metals and alloys are quite small (typically with a tensile elongation of a few percents). There are numerous ways to improve the ductility of the NC 304 SS. For instance one can enlarge the grain size of NC 304 SS (by introducing ultra-fine grains) to promote work hardening capability. We will look into various approaches to improve the tensile ductility of NC 304 SS in our future studies.

The strong surface effect of TEM thin sample on the defect dynamics during in-situ irradiation need to be addressed and discussed, especially at high temperature (500C).

Response: We have added the following paragraph (lines 167-176) to our revised manuscript to discuss the thin film effect:

It is worth mentioning that free surfaces of the thin foil may play some roles in defect evolution during *in situ* irradiation studies. As shown in Extended Data Figure 5, the CG-SS in situ irradiated to 5 dpa at 500 °C has similar but somewhat higher void swelling as compared to the same type of materials irradiated with neutrons. However, both CG-SS and NC-SS are evaluated under the same *in situ* irradiation conditions. The thickness of TEM specimens is similar in both cases, ~ 100 nm. Therefore, the surface effect on radiation induced defects would be comparable. Noted that NC-SS has an average grain size of ~ 45 nm, much smaller than the thickness of TEM film. The significantly improved radiation tolerance in NC-SS arises mainly from the high-density grain boundaries, which are dominating sinks for defects.

The error bar for the void swelling measurement need to be discussed and marked. The relationship and difference between the current methods and the bulk measurement need to be discussed as well.

Response: The error bar has been added to Extended Data Figure 5.

In general, the void swelling in neutron irradiated bulk materials are derived from the measurement of dimensional changes, as the neutron radiation induced size change is often large enough (visible to naked eyes) to be measured in bulk specimens. For the heavy ion or Helium irradiated specimens, the magnitude of dimensional change is often very small. A profilometer is often used to measure the step height changes and then back calculate swelling [1, 2]. Another widely used technique to measure small dimensional change is to use TEM [2, 3]. In the current study, for both in situ and ex situ experiments, we have used the TEM method to probe void swelling. The error bars are derived from calculating void size and density from multiple TEM micrographs.

Refs:

- [1] E. G. Fu, A. Misra, H. Wang, Lin Shao, and X. Zhang, Interface enabled defects reduction in helium ion irradiated Cu/V nanolayers, *Journal of Nuclear Materials* **407** (2010) 178–188.
- [2] C. Sun, S. Zheng, C. C. Wei, Y. Wu, L. Shao, Y. Yang, K. T. Hartwig, S. A. Maloy, S. J. Zinkle, T. R. Allen, H. Wang, and X. Zhang, Superior radiation-resistant nanoengineered austenitic 304L stainless steel for applications in extreme radiation environments, *Scientific Reports*, **5** (2015) 7801.
- [3] E. Getto, K. Sun, S. Taller, A. M. Monterrosa, Z. Jiao, G. S. Was, Methodology for determining void swelling at very high damage under ion irradiation, *Journal of Nuclear Materials* **477** (2016) 273-279.

Such information has been added to both the main text (lines 166-167) and the extended information (lines 729-737). The added information reads as:

Lines 166-167: The void swelling in this study is measured using the TEM technique⁵⁴. More information is discussed in the Extended Data Figure 5.

Lines 729-727: In general, the void swelling in neutron irradiated bulk materials are derived from the measurement of dimensional changes, as the neutron radiation induced size change is often large enough (visible to naked eyes) to be measured in bulk specimens. For the heavy ion or Helium irradiated specimens, the magnitude of dimensional change is often very small. A profilometer is often used to measure the step height changes and then back calculate swelling^{51,66}. Another widely used technique to measure small dimensional change is to use TEM^{51,54}. In the current study, for both in situ and ex situ experiments, the TEM method is used to probe void swelling. The error bars are derived from calculating void size and density from multiple TEM micrographs.

References:

51. Sun, C., Zheng, S., Wei, C. C., Wu, Y., Shao, L., Yang, Y., Hartwig, K. T., Maloy, S. A., Zinkle, S. J., Allen, T. R., Wang, H. & Zhang, X. Superior radiation-resistant nanoengineered austenitic 304L stainless steel for applications in extreme radiation environments. *Sci. Rep.* **5**, 7801 (2015).
54. Getto, E., Sun, K., Taller, S., Monterrosa, A. M., Jiao, Z., Was, G. S. Methodology for determining void swelling at very high damage under ion irradiation, *J. Nucl. Mater.* **477**, 273-279 (2016).
66. Fu, E. G., Misra, A., Wang, H., Shao, L., Zhang, X. Interface enabled defects reduction in helium ion irradiated Cu/V nanolayers, *J. Nucl. Mater.* **407**, 178-188 (2010).

Reviewer #4 (Remarks to the Author):

The authors have addressed all the questions and comments I raised. In my view, the manuscript can be accepted for publication at Nature Communications.

REVIEWERS' COMMENTS:

Reviewer #3 (Remarks to the Author):

The authors have addressed all my concerns. The current version of manuscript looks great, and will have high impact in the related fields. Hence I would like to recommend it for publication in Nature Communications.

The final comments of four referees are copied below. Based on their comments, we do not need to address any new concerns.

Reviewer #1 (Remarks to the Author):

The overall response is satisfactory. I appreciated the efforts to re-simulate defect interactions by including defect clustering into the picture. The manuscript is recommended to publish as it is. One reviewer pointed out that the approach is not innovative enough. It is true that there are so many previous reports already on boundary engineering. But the approach reported here is more realistic than others for industry scale up process and the finding on its superior stability under extreme high temperature is quite important to the field. Hence, its impact is appropriate for Nat. Comm.

Reviewer #2 (Remarks to the Author):

The changes made by the authors in response to the referee comments are satisfactory.

Reviewer #3 (Remarks to the Author):

The authors have addressed all my concerns. The current version of manuscript looks great, and will have high impact in the related fields. Hence I would like to recommend it for publication in Nature Communications.

Reviewer #4 (Remarks to the Author):

The authors have addressed all the questions and comments I raised. In my view, the manuscript can be accepted for publication at Nature Communications.